

# Channel selection method for hyperspectral atmospheric infrared sounder using AIRS data based on layering

**Shujie Chang[1, 2,3], Zheng Sheng[1,2], Huadong Du[1,2], Wei Ge[1,2] and Wei Zhang[1,2]**

[1] College of Meteorology and Oceanography, National University of Defense Technology, Nanjing, China

[2] Collaborative Innovation Center on Forecast and Evaluation of Meteorological Disasters, Nanjing University of Information Science and Technology, Nanjing, China

[3] South China Sea Institute for Marine Meteorology, Guangdong Ocean University, Zhanjiang, China

**Correspondence:** Zheng Sheng (19994035@sina.com)

**Abstract.** Because a satellite channel's ability to resolve hyperspectral data varies with height, an improved channel selection method is proposed based on information content. An effective channel selection scheme for a hyperspectral atmospheric infrared sounder using AIRS data based on layering is proposed. The results are as follows: (1) Using the improved method, the atmospheric retrievable index is more stable, the value reaching 0.54. The



distribution of the temperature weight function is more continuous,
more closely approximating that of the actual atmosphere; (2)
Statistical inversion comparison experiments show that the accuracy
of the retrieval temperature, using the improved channel selection
method in this paper, is consistent with that of 1Dvar channel
selection. In the near space layer especially, from 10 hPa to 0.02 hPa,
the accuracy of the retrieval temperature of our improved channel
selection method is evidently improved by about 1 K. In general, the
accuracy of the retrieval temperature of ICS is improved. Especially,
from 100 hPa to 0.01 hPa, the accuracy of ICS can be improved by
more than 11 %; (3) Statistical inversion comparison experiments in
four typical regions indicate that ICS in this paper is significantly
better than NCS and PCS in different regions and shows latitudinal
variations. Especially, from 100 hPa to 0.01 hPa, the accuracy of ICS
can be improved by 7% to 13%, which means the ICS method
selected in this paper is feasible and shows great promise for
applications.
**1 Introduction**
Since the successful launch of the first meteorological satellite,
TIROS in the 1960s, satellite detection technology has developed
rapidly. Meteorological satellites observe Earth's atmosphere from



space and are able to record data from regions which are otherwise
difficult to observe. Satellite data greatly enrich the content and
range of meteorological observations, and consequently, atmospheric
exploration technology and meteorological observations have taken
us to a new stage in our understanding of weather systems and
related phenomena (Fang, 2014). From the perspective of vertical
atmospheric detection, satellite instruments are developing rapidly.
In their infancy, the traditional infrared detection instruments for
detecting atmospheric temperature and moisture profiles, such as
TOVS (Smith et al., 1991) or HIRS in ATOVS (Chahine, 1972; Li et
al., 2000; Liu, 2007), usually employed filter spectrometry. Even
though such instruments have played an important role in improving
weather prediction, it is difficult to continue to build upon
improvements in terms of detection accuracy and vertical resolution
due to the limitation of low spectral resolution. By using this kind of
filter-based spectroscopic detection instrument, therefore, it is
difficult to meet today's needs in numerical weather prediction (Eyre
et al., 1993). To meet this challenge, a series of plans for the creation
of high-spectral resolution atmospheric detection instruments has
been executed in the United States and in Europe in recent years:
One example is the AIRS (Atmospheric Infrared Sounder) on the
Earth Observation System, "Aqua", launched on May 4, 2002 from


67 the United States. AIRS has 2378 spectral channels with subpoint at

68 13 km and a detection height from the ground of up to 65 km

69 (Aumann et al., 2003; Hoffmann and Alexander, 2009; Gong et al.,

70 2011). The United States and Europe, in 2010, also installed the

71 CRIS (Cross-track Infrared Sounder) and the IASI (Inter-Attractive

72 Atmospheric Sounding Interferometer) on polar-orbiting satellites.

73 China also attaches great importance to the development of such

74 advanced detection technologies. In the early 1990s, the National

75 Satellite Meteorological Center began to investigate the principles

76 and techniques of hyperspectral resolution atmospheric detection.

77 China's development of interferometric atmospheric vertical

78 detectors eventually led to the launch of Fengyun No. 3, on May 27,

79 2008, and Fengyun No. 4 on December 11, 2016, both of which

80 were equipped with infrared atmospheric detectors. How best to use

81 the hyperspectral resolution detection data obtained from these

82 instruments, to obtain reliable atmospheric temperature and humidity

83 profiles, is an active area of intense study in atmospheric inversion

84 theory.

85 Due to technical limitations, only a limited number of channels

86 could at first be built into the general satellite detection instrument.

87 In this case, channel selection generally involved controlling the

88 channel weight function by utilizing the spectral response





characteristics of the channel (such as the center frequency,
bandwidth). With the development of detection technology,
increasing numbers of hyperspectral detectors were carried on
meteorological satellites. Due to the large number of channels and
data supported by such instruments today (such as AIRS with 2378
channels and IASI with 8461 channels), it has proven extremely
cumbersome to store, transmit, and process such data. Moreover,
there is a close correlation between each channel, causing an
ill-posedness of the inversion, potentially compromising accuracy of
the retrieval product based on hyperspectral resolution data.
However, hyperspectral detectors have many channels and
provide real-time mode prediction systems with vast quantities of
data, which can significantly improve prediction accuracy. But, if all
the channels are used to retrieve data, the retrieval time considerably
increases. Even more problematic are the glut of information
produced, and the unsuitability of the calculations for real-time
forecasting. Concurrently, the computer processing power must be
large enough to meet the demands of all the channels simultaneously
within the forecast time. It is important to select a group of channels
that can provide as much information as possible from the thousands
of channels' observations to improve the calculation efficiency and
retrieval quality.



Many researchers have studied the channel selection algorithm.
Menke (1984) first chose channels using a data precision matrix
method. Aires et al. (1999) made the selection using the Jacobian
matrix, which has been widely used since then (Aires et al., 2002;
Rabier et al., 2010). Rodgers (2000) indicated that there are two
useful quantities in measuring the information provided by the
observation data: Shannon information content and degrees of
freedom. The concept of information capacity then became widely
used in satellite channel selection. In 2007, Xu (2007) compared the
Shannon information content with the relative entropy, analyzing the
information loss and information redundancy. In 2008, Du et al.
(2008) introduced the concept of the atmospheric retrievable index
(ARI) as a criterion for channel selection, and in 2010, Wakita et al.
(2010) produced a scheme for calculating the information content of
the various atmospheric parameters in remote sensing using
Bayesian estimation theory. Kuai et al. (2010) analyzed both the
Shannon information content and degrees of freedom in channel
selection when retrieving $CO_2$ concentrations using thermal infrared
remote sensing and indicated that 40 channels could contain 75% of
the information from the total of 1016 channels. Cyril et al. (2003)
proposed the optimal sensitivity profile method based on the
sensitivity of different atmospheric components. Lupu et al. (2012)





used degrees of freedom for signals (DFS) to estimate the amount of
information contained in observations in the context of observing
system experiments. In addition, the singular value decomposition
method has also been widely used for channel selection (Prunet et al.,
2010; Zhang et al., 2011; Wang et al., 2014). In 2017, Chang et al.
(2017) selected a new set of Infrared Atmospheric Sounding
Interferometer (IASI) channels using the channel score index (CSI).
Richardson et al. (2018) selected 75 from 853 channels using
information content analysis to retrieve the cloud optical depth,
cloud properties, and position.
Today's main methods for channel selection (such as the data
precision matrix method (Menke, 1984), singular value
decomposition method (Prunet et al., 2010; Zhang et al., 2011; Wang
et al., 2014), and the Jacobi method (Aires et al., 1999; Rabier et al.,
2010) use only the weight function to study appropriate numerical
methods, the use of which allows sensitive channels to be selected.
The above-mentioned studies also take into account the sensitivity of
each channel to atmospheric parameters during channel selection,
while ignoring factors that impact retrieval results. The accuracy of
retrieval results depends not only on the channel weight function but
also on the channel noise, background field, and the retrieval
algorithm.



Currently, information content is often employed in channel
selection. During retrieval, this method delivers the largest amount
of information for the selected channel combination (Rodgers, 1996;
Du et al., 2008; He et al., 2012; Richardson et al., 2018). Although
this method has made great breakthroughs in both theory and
practice, however, it does not take the sensitivity of different
channels at different heights into consideration. This paper uses the
atmospheric retrievable index (ARI) as the index, which is based on
information content (Du et al., 2008; Richardson et al. 2018).
Channel selection is made at different heights, and an effective
channel selection scheme is proposed which fully considers various
factors, including the influence of different channels on the retrieval
results at different heights. This ensures the best accuracy of the
retrieval product when using the selected channel. In addition,
statistical inversion comparison experiments are used to verify the
effectiveness of the method.

**2 Channel selection indicator and scheme**
**2.1 Channel selection indicator**
According to the concept of information content, the information
content contained in a selected channel of a hyperspectral instrument
can be described as H (Rodgers, 1996; Rabier et al., 2010). The final





expression of H is:

$$H = -\frac{1}{2} ln|\hat{S} S_a^{-1}|$$


$$= -\frac{1}{2} ln|(S_a - S_a K^T (K S_a K^T + S_\varepsilon)^{-1} K S_a) S_a^{-1}|, \qquad (1)$$

where $S_a$ is the error covariance matrix of the background or the
estimated value of atmospheric profile, $\hat{S}$ represents the observation
error covariance matrix of each hyperspectral detector channel,
$\hat{S} = (S_a - S_a K^T (K S_a K^T + S_\varepsilon)^{-1} K S_a)$ denotes the covariance
matrix after retrieval by hyperspectral data, K is the weight function
matrix, which comes from the selected channel in the hyperspectral
data with respect to a specific atmospheric profile parameter.
In order to describe the accuracy of the retrieval results visually
and quantitatively, the atmospheric retrievable index (ARI), p, (Du et
al., 2008) is defined as follows:

$$p = 1 - exp(\frac{1}{2n} ln|\hat{S} S_a^{-1}|), \qquad (2)$$

where $S_a$ is the error covariance matrix of the background or the
estimated value of the atmospheric profile, and $\hat{S}$ represents the
observation error covariance matrix of each hyperspectral detector



channel. Assuming that before and after retrieval, the ratio of the
root mean square error of each element in the atmospheric state
vector is 1-p, then $\left|\hat{S}S_a^{-1}\right| = (1 - p)^{2n}$ is derived. By inverting the
equation, the ARI that is p can be obtained in Eq. (2), which
indicates the relative portion of the error that is eliminated by
retrieval. In fact, before and after retrieval, the ratio of the root mean
square error of each element cannot be 1-p. Therefore, p defined by
Eq. (1) is actually an overall evaluation of the retrieval result.

## 207   2.2 Channel selection scheme

The principle of channel selection is to find the optimum channel
combination after numbering the channels. This combination will
make the information content, H, or the ARI defined in this paper as
large as possible, in order to maintain the highest possible accuracy
in the retrieval results.

Let there be M layers in the vertical direction of the atmosphere

and N satellite channels. Selecting n from N channels, there will be
$C_N^n$ combinations in each layer, leading $C_N^n$ calculations to get $C_N^n$
kinds of p results. Furthermore, under the maximum one p-value, the
corresponding channel combination is used as the optimum channel
combination; therefore, the entire atmosphere must be calculated
$M \cdot C_N^n$ times. However, the calculation $M \cdot C_N^n$ times will be


particularly large, which makes this approach impractical in
calculating p for all possible combinations. Therefore, it is necessary
to design an effective calculation scheme, and such a scheme, i.e., a
channel selection method, using iteration is proposed, called the
"sequential absorption method". The method's main function is to
select ("absorb") channels one by one, taking the channel with the
maximum value of p. Through n iterations, n channels can be
selected as the final channel combination. The steps are as follows:
(1) The expression of information content in a single channel:
First, we use only one channel for retrieval. A row vector, k, in the
weight function matrix, K, is a weight function corresponding to the
channel. A diagonal element, $s_\varepsilon \frac{\partial^2 \Omega}{\partial v^2}$, in the $S_\varepsilon$ matrix is the error
variance in the channel. After observation in this channel, the error
covariance matrix is:
$$\hat{S} = S_a - S_a k^T (s_\varepsilon + k S_a k^T)^{-1} k S_a. \tag{3}$$
It should be noted that $(s_\varepsilon + k S_a k^T)$ is a single value in Eq. (3),
so Eq. (3) can be converted to:
$$\hat{S} = \left(I - \frac{S_a k^T k}{(s_\varepsilon + k S_a k^T)}\right) S_a = \left(I - \frac{(k S_a)^T k}{(s_\varepsilon + k (k S_a)^T)}\right) S_a. \tag{4}$$
Substituting Eq. (4) into Eq. (2) gives:
$$p = 1 - \exp\left(\frac{1}{2n} ln\left(\left|I - \frac{(k S_a)^T k}{(s_\varepsilon + k (k S_a)^T)}\right|\right)\right). \tag{5}$$





(2) Simplification of Eq. (5) p matrix:
Since $S_a$ is a positive definite symmetric matrix, it can be
decomposed into $S_a = (S_a^{1/2})^T (S_a^{1/2})$ and $S_\varepsilon = (S_\varepsilon^{1/2})^T (S_\varepsilon^{1/2})$.

Define $R = S_\varepsilon^{1/2} K S_a^{1/2}$.                     (6)

The matrix R can then be regarded as a weight function matrix,
normalized by the observed error and pre-observation error. A row
vector of R, $r = s_\varepsilon^{-1/2} k S_a^{1/2}$, represents the normalized weight
function matrix of a single channel. Substituting r into Eq. (5) gives:

$p = 1 - \exp(\frac{1}{2n} ln\left(\left|I - \frac{rr^T}{1+r^T r}\right|\right))$.                     (7)

For arbitrary row vectors, a and b, using the matrix property
$\det(I + a^T b) = 1 + ba^T$, the new expression for p is:

$$p = 1 - \exp\left(\frac{1}{2n} ln\left(1 - \frac{r^T r}{1 + r^T r}\right)\right)$$

$= 1 - \exp\left(\frac{1}{2n} ln\left(\frac{1}{1+r^T r}\right)\right)$
$= 1 - \exp\left(-\frac{1}{2n} ln(1 + r^T r)\right)$.                     (8)

(3) Iteration in a single layer:





First, the iteration in a single layer requires the calculation of R.
According to $S_a$, $S_\varepsilon$ , K and Eq. (6), R, which is r corresponding to
all the selected channels, can be calculated. Second, using Eq. (8), p
of each candidate channel can be calculated. Moreover, the channel
corresponding to maximum p is the selected channel for this
iteration. After a channel has been selected, according to Eq. (3) we
can use $\hat{S}$ to get $S_a$ for the next iteration. Finally, channels which
are not selected during this iteration are used as the candidate
channels for the next iteration.
When selecting n from N channels, it is necessary to calculate
(N-n/2)n≈Nn p values, which is much smaller than $C_N^n$. Of course,
the combination selected by this method is not completely
equivalent to the channel combination corresponding to the optimum
value of $C_N^n$ p, but it still satisfies the optimum value in a certain
sense. In addition to its high computational efficiency by using this
method, another advantage is that all channels can be recorded in the
order in which they are selected. In the actual application, if $n'$
channels are needed, and $n' <$ n, we will not need to select the
channel again, but record the selected channel only.
(4) Iteration for different altitudes:
Because satellite channel sensitivity varies with height, repeating
the iterative process of step (3), selects the optimum channels at





different heights. Assuming there are M layers in the atmosphere and
selecting n from N channels, it is necessary to calculate $M \cdot (N -$
$n/2)n \approx M \cdot Nn$  p values, a much smaller number than  $M \cdot C_N^n$.

**2.3 Statistical inversion method**
The inversion method of the atmospheric temperature profile can be
summarized in two categories: statistical inversion and physical
inversion. Statistical inversion is essentially a linear regression
model which uses a large number of satellite measurements and
atmospheric parameters to match samples and calculate their
correlation coefficient. Then, based on the correlation coefficient, the
required parameters of the independent measurements obtained by
the satellite are retrieved. Because the method does not directly solve
the radiation transfer equation, it has the advantages of fast
calculation speed. In addition, the solution is stable, which makes it
one of the highest precision methods (Chedin et al., 1985). Therefore,
the statistical inversion method will be used for our channel
selection experiment and a regression equation will be established.

According to an empirical orthogonal function, the atmospheric

temperature (or humidity), T, and the bright temperature,  $T_b$, are
expanded thus:





$\mathrm{T} = T^* \cdot A,$          (9)

$T_b = T_b^* \cdot A,$          (10)

where $T^*$ and $T_b^*$ are the eigenvectors of the covariance matrix of
temperature (or humidity) and brightness temperature, respectively.
A and B stand for the corresponding expansion coefficient vectors of
temperature (humidity) and brightness temperature.
Using the least squares method and the orthogonal property, the
coefficient conversion matrix, V, is introduced:

$\mathrm{A} = \mathrm{V} \cdot B,$          (11)

where $V = AB^T (BB^T)^{-1}.$          (12)

Using the orthogonality, we get:

$\mathrm{B} = (T_b^*)^T T_b,$          (13)

$\mathrm{A} = (T^*)^T T.$          (14)

For convenience, the anomalies of the state vector (atmospheric





temperature), T, and the observation vector (bright temperature), $T_b$,
are taken:

$$\widehat{T} = \overline{T} + \widehat{T}^{'} = \overline{T} + GT_b^{'} = \overline{T} + G(T_b - \overline{T_b}),$$   (15)

where $\overline{T}$ and $\overline{T_b}$ are the corresponding average values of the
elements, respectively. $T^{'}$ and $T_b^{'}$ represent the corresponding
anomalies of the elements, respectively.

Assuming there are k sets of observations, a sample anomaly

matrix with k vectors can be constructed:

$$T^{'} = (t_1^{'}, \ t_2^{'}, \ \cdots, \ t_k^{'}),$$   (16)

$$T_b^{'} = (t_{b1}^{'}, \ t_{b2}^{'}, \ \cdots, \ t_{bk}^{'}).$$   (17)

Define the inversion error matrix as:


$$\delta = \overline{T} - \widehat{T} = \widehat{T}^{'} - T^{'}.$$   (18)


The retrieval error covariance matrix is:


$$S_\delta = \frac{1}{k - n - 1} \delta \delta^{T}$$



$\qquad = \frac{1}{k-n-1}(T^{'} - GT_{b}^{'})(T^{'} - GT_{b}^{'})^{T}$
$\qquad = \frac{k-1}{k-n-1}(S_{e} - G^{T}S_{xy} - S_{xy}G^{T} + GS_{y}G^{T}),$ (19)

where

$S_{e} = \frac{1}{k-1}T^{'} \; T^{'\; T}$ ,
$S_{y} = \frac{1}{k-1}T_{b}^{'} \; T_{b}^{'\; T}$ ,
$S_{xy} = \frac{1}{k-1}T^{'} \; T_{b}^{'\; T}$ . (20)

$\quad$ $S_{e}$ stands for the sample covariance matrix of T, $S_{y}$ denotes the
sample covariance matrix of $T_{b}$, and $S_{xy}$ represents the covariance
matrix of T and $T_{b}$. The elements on the diagonal of the error
covariance matrix, $S_{\delta}$, represent the retrieval error variance of T.
The matrix G that minimizes the overall error variance is the least
squares coefficient matrix of the regression equation (15), which
meets the criteria:

$\delta^{2} = tr(S_{\delta}) = min.$ (21)

$\quad$ Equation (21) takes a derivative with respect to G, $\frac{\partial}{\partial G}tr(S_{\delta}) =$
$0 = (-2S_{xy} + 2GS_{y})$, which means that:






$$G = S_{xy}S_y^{-1}. \hspace{4cm}(22)$$

Substituting Eq. (22) into Eq. (15) finally gives the least squares
solution as:

$$\widehat{T} = \overline{T} + S_{xy}S_y^{-1}(T_b - \overline{T_b}). \hspace{2.5cm}(23)$$

It should be noted that the least squares solution obtained here
aims to minimize the sum of the error variance for each element in
the atmospheric state vector after retrieval of observations has been
completed several times. At present, statistical multiple regression is
widely used in the retrieval of atmospheric profiles based on
atmospheric remote sensing data. As long as there are enough data,
$S_{xy}$ and $S_y$ can be determined.

**3. Channel selection experiment**
**3.1 Data and model**
The Atmospheric Infrared Sounder (AIRS) instrument suite is
designed to measure the Earth's atmospheric water vapor and
temperature profiles on a global scale. AIRS is a continuously
operating cross-track scanning sounder, consisting of a telescope that



feeds an echelle spectrometer. The AIRS infrared spectrometer
acquires 2378 spectral samples at a resolution $\lambda/\Delta\lambda$, ranging from
1086 to 1570, in three bands: 3.74 μm to 4.61 μm, 6.20 μm to 8.22
μm, and 8.8 μm to 15.4 μm. The spatial footprint of the infrared
channels is 1.1° in diameter, which corresponds to about $15\times15$ km
at the nadir. The spectral range includes 4.2 μm for important
temperature detection, 15 μm for $CO_2$, 6.3 μm for water vapor, and
9.6 μm for ozone absorption bands. The absolute accuracy of the
measured radiation is better than 0.2 K. Moreover, global
atmospheric profiles can be detected every day, and the four imaging
channels of visible/near infrared are always filled. Due to radiometer
noise and faults, there are currently only 2047 effective channels.
However, compared with previous infrared detectors, AIRS boasts a
significant improvement in both the number of channels and spectral
resolution (Aumann, 1994; Huang et al., 2005; Li et al., 2005).

AIRS provides real-time mode prediction systems with vast

quantities of data, which greatly improves prediction accuracy.
However, if all the channels are used to retrieve data, the retrieval
time becomes greatly extended. Even more problematic are the huge
amounts of information and calculations not being suitable for
real-time forecasting.

The root mean square error of an AIRS infrared channel is shown

in Fig. 1, with black spots, indicating that not all the instrument
channels possess a measurement error of less than 0.2 K. There are a
few with extremely large measurement errors, which reduce the
accuracy of prediction to some extent. Moreover, not all channels
possess the same measurement error. At present, more than 300
channels have not been used because their errors exceed 1 K. If data
from these channels were to be used for retrieval, the accuracy of the
retrieval could be reduced. Therefore, it is necessary to select a
group of channels to improve the calculation efficiency and retrieval
quality. In this paper we study channel selection for temperature
profile retrieval by AIRS.

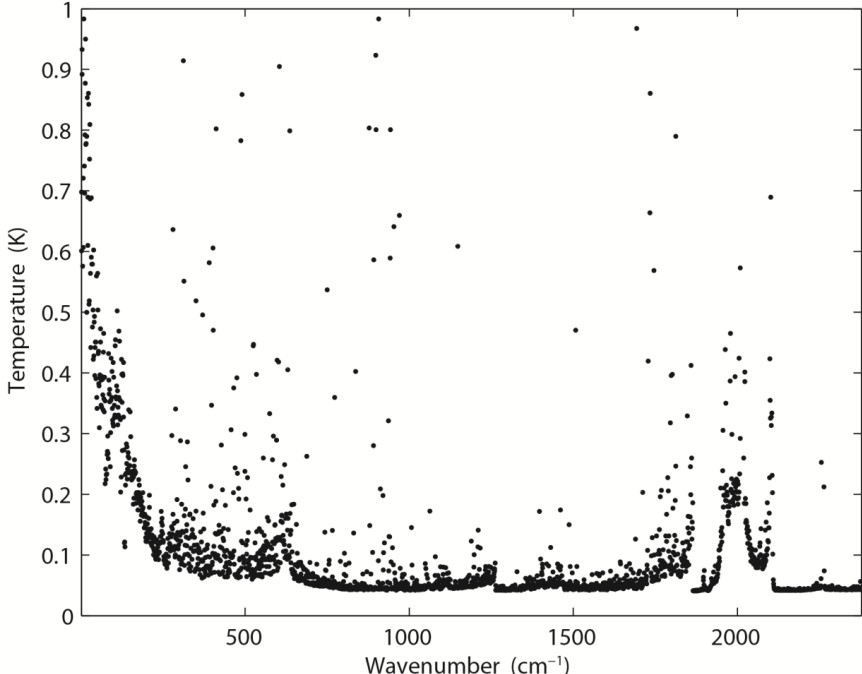


**Figure 1.** Root mean square error of AIRS infrared channel (black





spots).

For the radiative transfer model and its weight function matrix, K,
the RTTOV v12 fast radiative transfer model is used. RTTOV is an
evolution of RTTOV v11, adding and upgrading many features. The
model allows rapid simulations (1 ms for 40 channel ATOVS on a
desktop PC) of radiances for satellite visible, infrared, or microwave
nadir scanning radiometers given atmospheric profiles of
temperature and variable gas concentration, and cloud and surface
properties. The only mandatory gas included as a variable for
RTTOV v12 is water vapor. Optionally, ozone, carbon dioxide,
nitrous oxide, methane, carbon monoxide, and sulfur dioxide can be
included, with all other constituents assumed to be constant. RTTOV
v12 can accept input profiles on any defined set of pressure levels.
The majority of RTTOV v12 coefficient files are based on the 54
levels shown in Table 1, ranking from 1050 hPa to 0.01 hPa, though
coefficients for some hyperspectral sounders are also available on
101 levels.

**Table 1.** Pressure levels adopted for RTTOV v12 54 pressure level
coefficients and profile limits within which the transmittance
calculations are valid. Note that the gas units here are ppmv.


(From https://www.nwpsaf.eu/site/software/rttov/, RTTOV Users
guide, 2019).

| Level | Pressure | Tmax | Tmin | Qmax | Qmin | $Q_2$max | $Q_2$min | $Q_2$Ref |
|-------|----------|------|------|------|------|----------|----------|----------|
| Number | hPa | K | K | ppmv* | ppmv* | ppmv* | ppmv* | ppmv* |
| 1 | 0.01 | 245.95 | 143.66 | 5.24 | 0.91 | 1.404 | 0.014 | 0.296 |
| 2 | 0.01 | 252.13 | 154.19 | 6.03 | 1.08 | 1.410 | 0.069 | 0.321 |
| 3 | 0.03 | 263.71 | 168.42 | 7.42 | 1.35 | 1.496 | 0.108 | 0.361 |
| 4 | 0.03 | 280.12 | 180.18 | 8.10 | 1.58 | 1.670 | 0.171 | 0.527 |
| 5 | 0.13 | 299.05 | 194.48 | 8.44 | 1.80 | 2.064 | 0.228 | 0.769 |
| 6 | 0.23 | 318.64 | 206.21 | 8.59 | 1.99 | 2.365 | 0.355 | 1.074 |
| 7 | 0.41 | 336.24 | 205.66 | 8.58 | 2.49 | 2.718 | 0.553 | 1.471 |
| 8 | 0.67 | 342.08 | 197.17 | 8.34 | 3.01 | 3.565 | 0.731 | 1.991 |
| 9 | 1.08 | 340.84 | 189.50 | 8.07 | 3.30 | 5.333 | 0.716 | 2.787 |
| 10 | 1.67 | 334.68 | 179.27 | 7.89 | 3.20 | 7.314 | 0.643 | 3.756 |
| 11 | 2.50 | 322.5 | 17627 | 7.75 | 2.92 | 9.191 | 0.504 | 4.864 |
| 12 | 3.65 | 312.51 | 175.04 | 7.69 | 2.83 | 10.447 | 0.745 | 5.953 |
| 13 | 5.19 | 303.89 | 173.07 | 7.58 | 2.70 | 12.336 | 1.586 | 6.763 |
| 14 | 7.22 | 295.48 | 168.38 | 7.53 | 2.54 | 12.936 | 1.879 | 7.109 |
| 15 | 9.84 | 293.33 | 166.30 | 7.36 | 2.46 | 12.744 | 1.322 | 7.060 |
| 16 | 13.17 | 287.05 | 16347 | 7.20 | 2.42 | 11.960 | 0.719 | 6.574 |
| 17 | 17.33 | 283.36 | 161.49 | 6.96 | 2.20 | 11.105 | 0.428 | 5.687 |
| 18 | 22.46 | 280.93 | 161.47 | 6.75 | 1.71 | 9.796 | 0.278 | 4.705 |
| 19 | 28.69 | 282.67 | 162.09 | 6.46 | 1.52 | 8.736 | 0.164 | 3.870 |
| 20 | 36.17 | 27993 | 162.49 | 6.14 | 1.31 | 7.374 | 0.107 | 3.111 |
| 21 | 45.04 | 27315 | 164.66 | 5.90 | 1.36 | 6.799 | 0.055 | 2.478 |
| 22 | 55.44 | 265.93 | 166.19 | 6.21 | 1.30 | 5.710 | 0.048 | 1.907 |
| 23 | 67.51 | 264.7 | 167.42 | 9.17 | 1.16 | 4.786 | 0.043 | 1.440 |
| 24 | 81.37 | 261.95 | 159.98 | 17.89 | 0.36 | 4.390 | 0.038 | 1.020 |
| 25 | 97.15 | 262.43 | 163.95 | 20.30 | 0.01 | 3.619 | 0.016 | 0.733 |





| | | | | | | | |
|---|---|---|---|---|---|---|---|
| 26 | 114.94 | 259.57 | 168.59 | 33.56 | 0.01 | 2.977 | 0.016 | 0.604 |
| 27 | 134.83 | 259.26 | 169.71 | 102.24 | 0.01 | 2.665 | 0.016 | 0.489 |
| 28 | 156.88 | 260.13 | 169.42 | 285.00 | 0.01 | 2.351 | 0.013 | 0.388 |
| 29 | 181.14 | 262.27 | 17063 | 714.60 | 0.01 | 1.973 | 0.010 | 0.284 |
| 30 | 207.61 | 264.45 | 174.11 | 1464.00 | 0.01 | 1.481 | 0.013 | 0.196 |
| 31 | 236.28 | 270.09 | 177.12 | 2475.60 | 0.01 | 1.075 | 0.016 | 0.145 |
| 32 | 267.10 | 277.93 | 181.98 | 4381.20 | 0.01 | 0.774 | 0.015 | 0.110 |
| 33 | 300.00 | 285.18 | 184.76 | 6631.20 | 0.01 | 0.628 | 0.015 | 0.086 |
| 34 | 334.86 | 293.68 | 187.69 | 9450.00 | 1.29 | 0.550 | 0.016 | 0.073 |
| 35 | 371.55 | 300.12 | 190.34 | 12432.00 | 1.52 | 0.447 | 0.015 | 0.063 |
| 36 | 409.89 | 302.63 | 194.40 | 15468.00 | 2.12 | 0.361 | 0.015 | 0.057 |
| 37 | 449.67 | 304.43 | 198.46 | 18564.00 | 2.36 | 0.284 | 0.015 | 0.054 |
| 38 | 490.&5 | 307.2 | 201.53 | 21684.00 | 2.91 | 0.247 | 0.015 | 0.052 |
| 39 | 532.56 | 31217 | 202.74 | 24696.00 | 3.67 | 0.199 | 0.015 | 0.050 |
| 40 | 572.15 | 31556 | 201.61 | 27480.00 | 3.81 | 0.191 | 0.012 | 0.050 |
| 41 | 618.07 | 318.26 | 189.95 | 30288.00 | 6.82 | 0.171 | 0.010 | 0.049 |
| 42 | 661.00 | 321.71 | 189.95 | 32796.00 | 6.07 | 0.128 | 0.009 | 0.048 |
| 43 | 703.59 | 327.95 | 189.95 | 55328.00 | 6.73 | 0.124 | 0.009 | 0.047 |
| 44 | 745.48 | 333.77 | 189.95 | 37692.00 | 8.71 | 0.117 | 0.009 | 0.046 |
| 45 | 786.33 | 336.46 | 189.95 | 39984.00 | 8.26 | 0.115 | 0.008 | 0.045 |
| 46 | 825.75 | 338.54 | 189.95 | 42192.00 | 7.87 | 0.113 | 0.008 | 0.043 |
| 47 | 863.40 | 342.55 | 189.95 | 44220.00 | 7.53 | 0.111 | 0.007 | 0.041 |
| 48 | 898.93 | 346.23 | 189.95 | 46272.00 | 7.23 | 0.108 | 0.006 | 0.040 |
| 49 | 931.99 | 34924 | 189.95 | 47736.00 | 6.97 | 0.102 | 0.006 | 0.038 |
| 50 | 962.26 | 349.92 | 189.95 | 51264.00 | 6.75 | 0.099 | 0.006 | 0.034 |
| 51 | 989.45 | 350.09 | 189.95 | 49716.00 | 6.57 | 0.099 | 0.006 | 0.030 |
| 52 | 1013.29 | 360.09 | 189.95 | 47208.00 | 6.41 | 0.094 | 0.006 | 0.028 |
| 53 | 1033.54 | 350.09 | 189.95 | 47806.00 | 6.29 | 0.094 | 0.006 | 0.027 |
| 54 | 1050.00 | 350.09 | 189.95 | 47640.00 | 6.19 | 0.094 | 0.006 | 0.027 |




The weight function matrix, K (Jacobian matrix), in this paper is
the weight function matrix of the atmospheric characteristics. In
order to correspond to the selected profiles, the atmosphere is
divided into 137 layers, each of which contains corresponding
atmospheric characteristics, such as temperature, pressure, and the
humidity distribution. Each element in the weight function matrix
can be written as $\partial yi/\partial xj$. The subscript i is used to identify the
satellite channel, and the subscript j is used to identify the
atmospheric characteristics. Therefore, $\partial yi/\partial xj$ indicates the variation
in radiation brightness temperature in a given satellite channel, when
a given atmospheric characteristic in a given layer changes. We are
thus able to establish which layer of the satellite channel is
particularly sensitive to which atmospheric characteristic
(temperature, various gas contents) in the vertical atmosphere. The
RTTOV_K (the K mode), is used to calculate the matrix H(X0) for a
given atmospheric profile characteristic.

**3.2 Channel selection comparison experiment and results**
In order to verify the effectiveness of the method, three sets of
comparison experiments were conducted. First, 324 channels used
by the EUMETSAT Satellite Application Facility on Numerical
Weather Prediction (NWP SAF) were selected. NCS is short for



NWP channel selection in this paper. The products were released by
the NWPSAF 1DVar (one-dimensional variational analysis) scheme,
in accordance with the requirements of the NWPSAF. Second, 324
channels were selected using the information capacity method. This
method was adopted by Du et al. (2008) without the consideration of
layering. PCS is short for primary channel selection in this paper.
Third, 324×M channels were selected using the information
capacity method for the M layer atmosphere. ICS is short for
improved channel selectionin this paper. In order to verify the
retrieval effectiveness after channel selection, statistical inversion
comparison experiments were performed using 5000 temperature
profiles provided by the ECMWF dataset, which will be introduced
in Sect. 4.
The observation error covariance matrix, $S_\varepsilon$, in the experiment is
provided by NWP SAF 1Dvar. In general, it can be converted to a
diagonal matrix, the elements of which are the observation error
standard deviation of each hyperspectral detector channel, which is
the square of the root mean square error for each channel. The root
mean square error of an AIRS infrared channel is shown in Fig. 1.
The error covariance matrix of the background, $S_a$, is calculated
using 5000 samples of the IFS-137 data provided by the ECMWF
dataset (download address:





495 https://www.nwpsaf.eu/site/update-137-level-nwp-profile-dataset/,

496 2019). The covariance matrix of temperature is shown in Fig. 2, the

497 results are consistent with the previous study by Du et al. (2008).


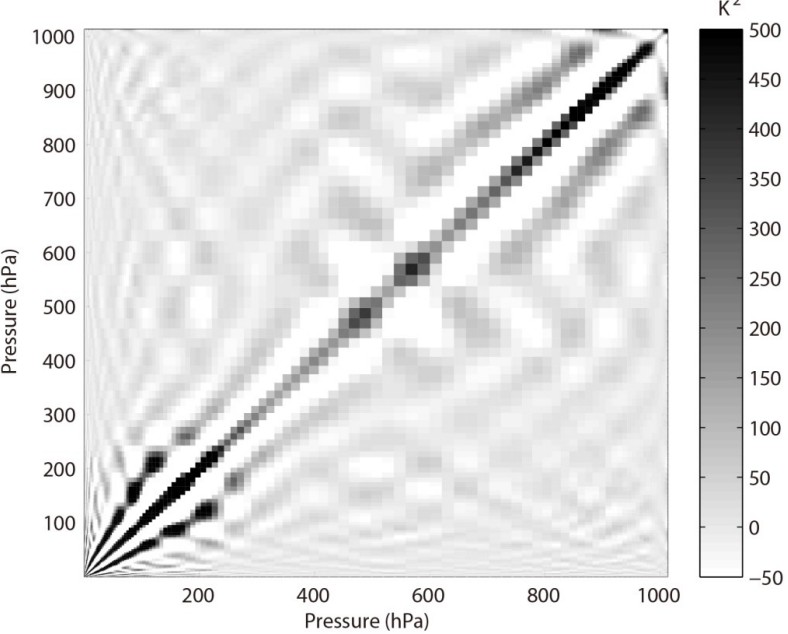


500 **Figure 2.** Error covariance matrix of temperature (shaded).


502   The reference atmospheric profiles are from the IFS-137 database,

503 and the temperature weight function matrix is calculated using the

504 RTTOV_K mode, as shown in Fig. 3; the results are consistent with

505 those of the previous study by Du et al. (2008). For the air-based

506 passive atmospheric remote sensing studied in this paper, when the

507 same channel detects the atmosphere from different observation



angles, the value of the weight function matrix K changes due to the
limb effect. Therefore, when we select channels, the results differ
because of the different observation angles. But due to the selection
principle and method are exactly the same and our key is the
selection method; we do not discuss, therefore, the variation in
observation angle when making a selection.

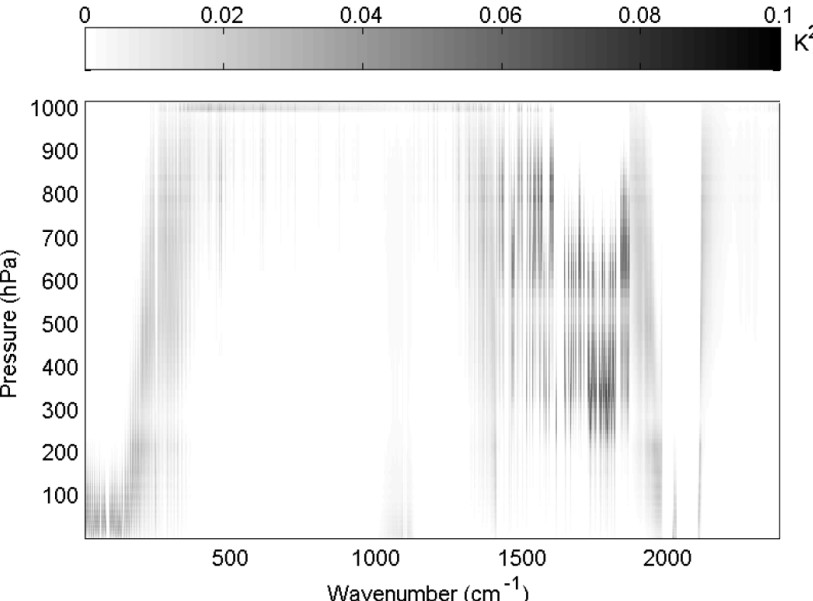


**Figure 3.** Temperature weight function matrix (shaded).

In order to verify the effectiveness of ICS, the distribution of 324
channels, without considering layering, in the AIRS bright
temperature spectrum is indicated in Fig. 4. The background



brightness temperature is the simulated AIRS observation brightness
temperature, which is from the atmospheric profile in RTTOV put
into the model. Figure 4(a) shows the 324 channels selected by PCS,
while Fig. 4(b) shows the 324 channels selected by NCS.



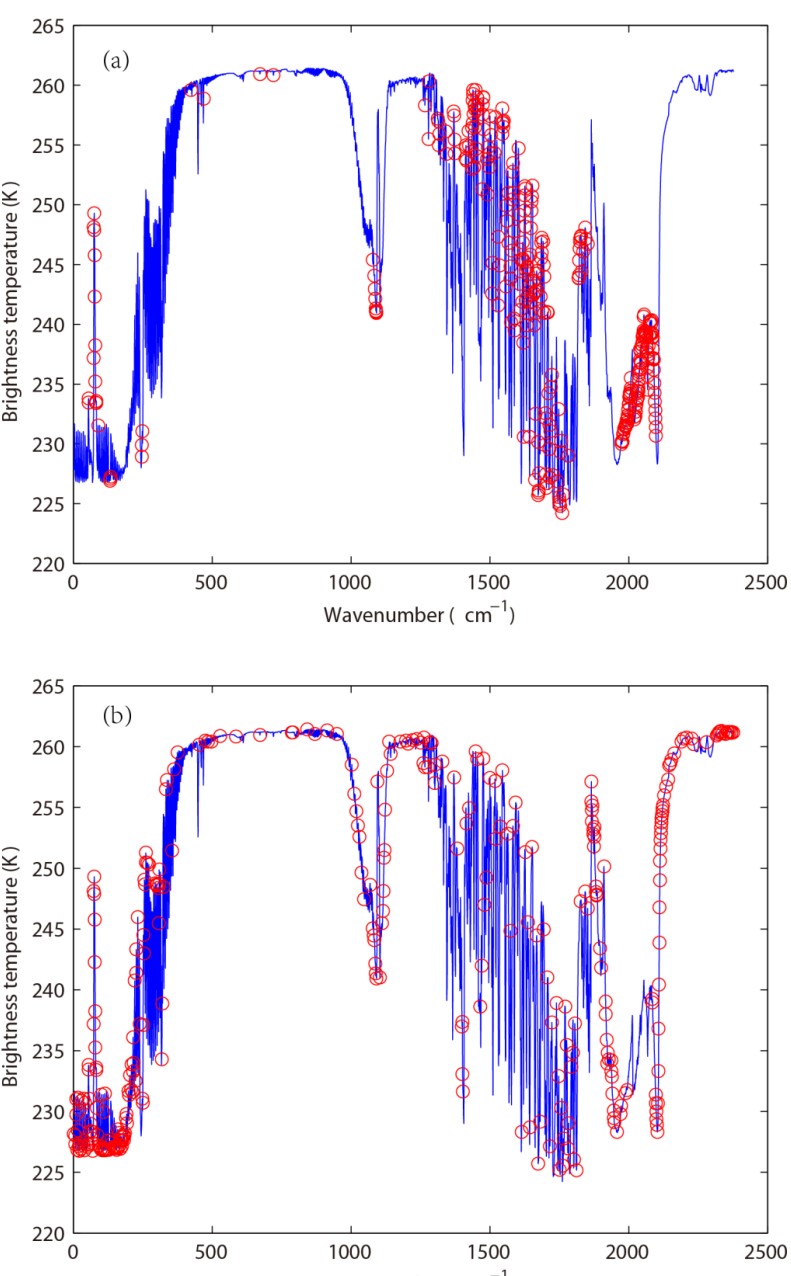


**Figure 4.** The distribution of different channel selection methods

without considering layering in the AIRS bright temperature



spectrum (blue line). (a) 324 channels selected by PCS (red circles).
(b) 324 channels selected by NCS (red circles).
Without considering layering, the main differences between the
324 channels selected by PCS and NCS are as follows: (1) When the
wavenumber approaches 1000, the wavelength is 11 μm (1/1000).
Near this band, fewer channels are selected by PCS because the
retrieval of ground temperature is considered by NCS; (2) When the
wavenumber is near 1200, the wavelength is 9 μm (1/1200). Near
this band, no channels are selected by PCS because the retrieval of
$O_3$ is not considered in this paper; (3) When the wavenumber
approaches 1500, the wavelength is 6.7 μm (1/1500). As is known,
the spectral range from 6 μm to 7 μm corresponds to water vapor
absorption bands, but fewer channels are selected by NCS; (4) When
the wavenumber is close to 2000, it derives a wavelength of 5 μm
(1/2000), which includes 4.2 μm for $N_2O$ and 4.3 μm for $CO_2$
absorption bands. As is shown in Fig. 4, fewer channels are selected
by PCS in those bands. PCS is favorable for atmospheric
temperature detection in the high temperature zone; (5) In the near
infrared area, the wavenumber exceeds 2200, deriving a wavelength
of less than 4 μm (1/2000). A small number of channels is selected
by NCS, but no channels are selected by PCS.
Above all, the information content used in this paper only takes





the temperature profile retrieval into consideration, so the channel
combination of PCS is inferior to that of NCS for the retrieval of
surface temperature and the $O_3$ profile. The advantages of the
channel selection method based on information content in this paper
are mainly reflected in: (1) Near space (20–100 km) is less affected
by the ground surface, so the retrieval result of PCS is better than
that of NCS. (2) Due to the method selected in this paper there are
more channels at 4.2 µm for $N_2O$ and 4.3 µm for $CO_2$ absorption
bands; the channel combination of PCS is superior to that of NCS
for atmospheric temperature detection in the high temperature zone.

By comparing channel selection without considering layering,

we note the general advantages and disadvantages of PCS and NCS
for the retrieval of atmosphere and can improve the channel
selection scheme. First, the retrieval of the temperature profile for
324 channels selected by PCS is obtained. The relationship between
the number of iterations and the ARI is shown in Fig. 5.

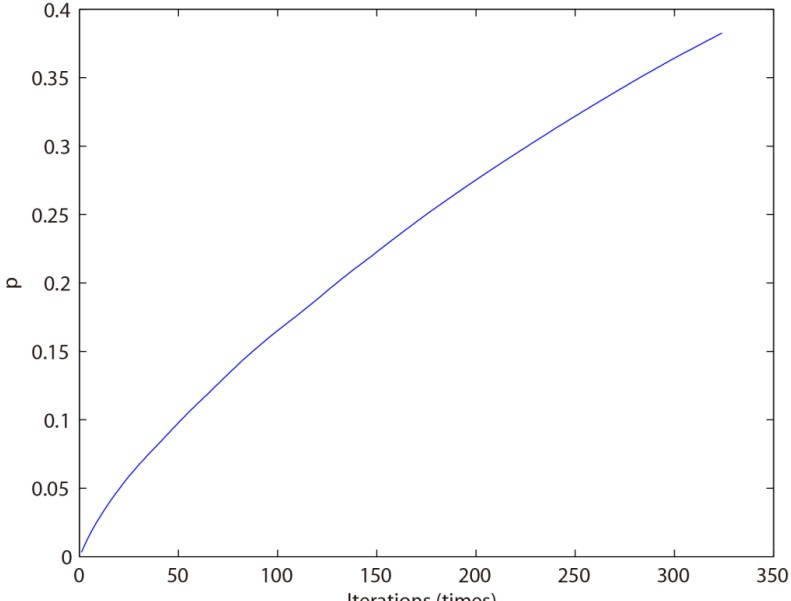


**Figure 5.** The relationship between the number of iterations and ARI

for PCS.


The ARI tends to be 0.38 and is not convergent, so the PCS

method needs to be improved. In this paper, the atmosphere is

divided into 137 layers, and based on the information content and

iteration, 324 channels are selected for each layer. Moreover, the

temperature profile of each layer can be retrieved. The relationship

between the number of iterations and the ARI is shown in Fig. 6.

When the number of iterations approaches 100, the ARI of ICS tends

to be stable, reaching 0.54. Thus, in terms of the ARI and

convergence, the ICS method is superior to that of PCS.



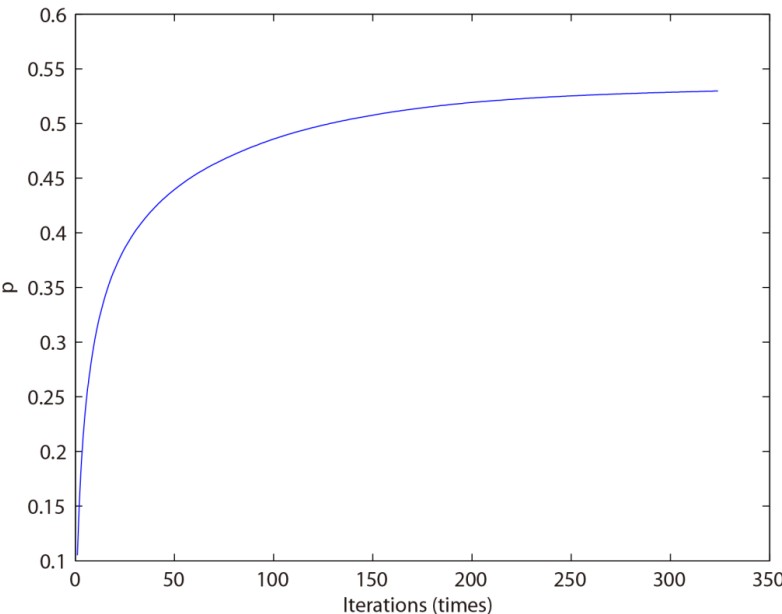

**Figure 6.** The relationship between the number of iterations and the

ARI for ICS.

Furthermore, because an iterative method is used to select

channels, the order of each selected channel is determined by the

contribution from the ARI. The weight function matrix of the top

324 selected channels, according to channel order, is shown in Fig.

7.

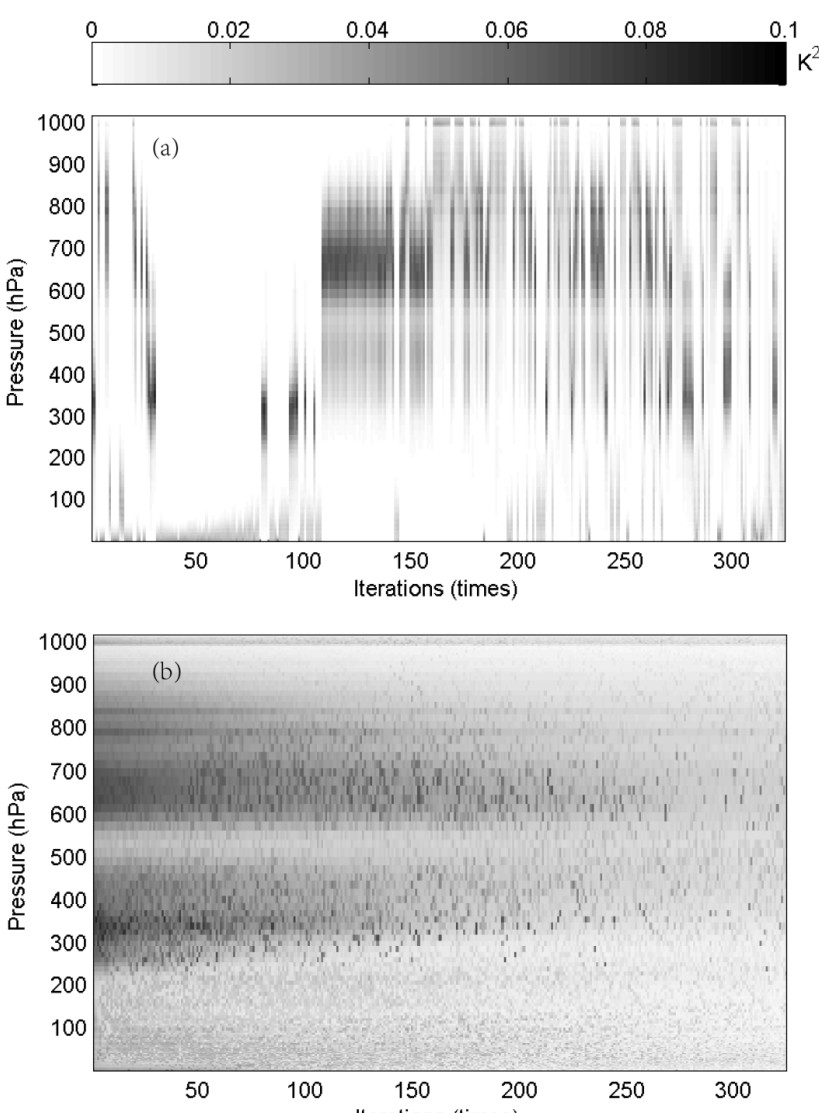


**Figure 7.** The relationship between the number of iterations and the

weight function of the top 324 selected channels (shaded). (a) PCS.

(b) ICS.






As illustrated in Fig. 7, in the first 100 iterations, the distribution
of the temperature weight function for PCS is relatively scattered; it
does not reflect continuity between the adjacent layers of the
atmosphere. Besides, the ICS result is better than that of PCS,
showing that: (1) the distribution of the temperature weight function
is more continuous and reflects the continuity between adjacent
layers of the atmosphere; (2) regardless of the number of iterations,
the maximum value of the weight function is stable near 300–400
hPa and 600–700 hPa, without scattering, which resembles more
closely the scenario in real atmosphere.

**4. Statistical multiple regression experiment**
**4.1 Temperature profile database**
A new database including a representative collection of 25,000
atmospheric profiles from the European Centre for Medium-range
Weather Forecasts (ECMWF) was used. The profiles were given in a
137-level vertical grid extending from the surface up to 0.01 hPa.
The database was divided into five subsets focusing on diverse
sampling characteristics such as temperature, specific humidity,
ozone mixing ratio, cloud condensates, and precipitation. In contrast
with earlier releases of the ECMWF diverse profile database, the
137-level database places greater emphasis on preserving the





statistical properties of sampled distributions produced by the
Integrated Forecasting System (IFS). IFS-137 spans the period from
September 1, 2013 to August 31, 2014. There are two operational
analyses each day (at 00z and 12z), and the modeling grid contains
2,140,702 grid points. The pressure levels adopted for IFS-137 are
shown in Table 2.

**Table 2.** Pressure levels adopted for IFS-137 137 pressure levels (in
hPa).

| Level number | pressure hPa | Level number | pressure hPa | Level number | pressure hPa | Level number | pressure hPa | Level number | pressure hPa |
|---|---|---|---|---|---|---|---|---|---|
| 1 | 0.02 | 31 | 12.8561 | 61 | 106.4153 | 91 | 424.019 | 121 | 934.7666 |
| 2 | 0.031 | 32 | 14.2377 | 62 | 112.0681 | 92 | 441.5395 | 122 | 943.1399 |
| 3 | 0.0467 | 33 | 15.7162 | 63 | 117.9714 | 93 | 459.6321 | 123 | 950.9082 |
| 4 | 0.0683 | 34 | 17.2945 | 64 | 124.1337 | 94 | 478.3096 | 124 | 958.1037 |
| 5 | 0.0975 | 35 | 18.9752 | 65 | 130.5637 | 95 | 497.5845 | 125 | 964.7584 |
| 6 | 0.1361 | 36 | 20.761 | 66 | 137.2703 | 96 | 517.4198 | 126 | 970.9046 |
| 7 | 0.1861 | 37 | 22.6543 | 67 | 144.2624 | 97 | 537.7195 | 127 | 976.5737 |
| 8 | 0.2499 | 38 | 24.6577 | 68 | 151.5493 | 98 | 558.343 | 128 | 981.7968 |
| 9 | 0.3299 | 39 | 26.7735 | 69 | 159.1403 | 99 | 579.1926 | 129 | 986.6036 |
| 10 | 0.4288 | 40 | 29.0039 | 70 | 167.045 | 100 | 600.1668 | 130 | 991.023 |
| 11 | 0.5496 | 41 | 31.3512 | 71 | 175.2731 | 101 | 621.1624 | 131 | 995.0824 |
| 12 | 0.6952 | 42 | 33.8174 | 72 | 183.8344 | 102 | 642.0764 | 132 | 998.8081 |
| 13 | 0.869 | 43 | 36.4047 | 73 | 192.7389 | 103 | 662.8084 | 133 | 1002.225 |
| 14 | 1.0742 | 44 | 39.1149 | 74 | 201.9969 | 104 | 683.262 | 134 | 1005.356 |
| 15 | 1.3143 | 45 | 41.9493 | 75 | 211.6186 | 105 | 703.3467 | 135 | 1008.224 |
| 16 | 1.5928 | 46 | 44.9082 | 76 | 221.6146 | 106 | 722.9795 | 136 | 1010.849 |
| 17 | 1.9134 | 47 | 47.9915 | 77 | 231.9954 | 107 | 742.0855 | 137 | 1013.25 |
| 18 | 2.2797 | 48 | 51.199 | 78 | 242.7719 | 108 | 760.5996 | | |
| 19 | 2.6954 | 49 | 54.5299 | 79 | 253.9549 | 109 | 778.4661 | | |
| 20 | 3.1642 | 50 | 57.9834 | 80 | 265.5556 | 110 | 795.6396 | | |
| 21 | 3.6898 | 51 | 61.5607 | 81 | 277.5852 | 111 | 812.0847 | | |
| 22 | 4.2759 | 52 | 65.2695 | 82 | 290.0548 | 112 | 827.7756 | | |
| 23 | 4.9262 | 53 | 69.1187 | 83 | 302.9762 | 113 | 842.6959 | | |





| 24 | 5.6441 | 54 | 73.1187 | 84 | 316.3607 | 114 | 856.8376 |
| 25 | 6.4334 | 55 | 77.281 | 85 | 330.2202 | 115 | 870.2004 |
| 26 | 7.2974 | 56 | 81.6182 | 86 | 344.5663 | 116 | 882.791 |
| 27 | 8.2397 | 57 | 86.145 | 87 | 359.4111 | 117 | 894.6222 |
| 28 | 9.2634 | 58 | 90.8774 | 88 | 374.7666 | 118 | 905.7116 |
| 29 | 10.372 | 59 | 95.828 | 89 | 390.645 | 119 | 916.0815 |
| 30 | 11.5685 | 60 | 101.0047 | 90 | 407.0583 | 120 | 925.7571 |

The locations of selected profiles of temperature, specific
humidity, and cloud condensate subsets of the IFS-91 and IFS-137
databases are plotted on the map in Fig. 8. In the IFS-91 database,
the sampling is fully determined by the selection algorithm, which
makes the geographical distributions very inhomogeneous. Selected
profiles represent those regions where gradients of the sampled
variable are the strongest: in the case of temperature, mid- and
high-latitudes dominate, while humidity and cloud condensate
subsets concentrate at low latitudes. However, the IFS-137 database
shows a much more homogeneous spatial distribution in all the
sampling subsets, which is a consequence of the randomized
selection.

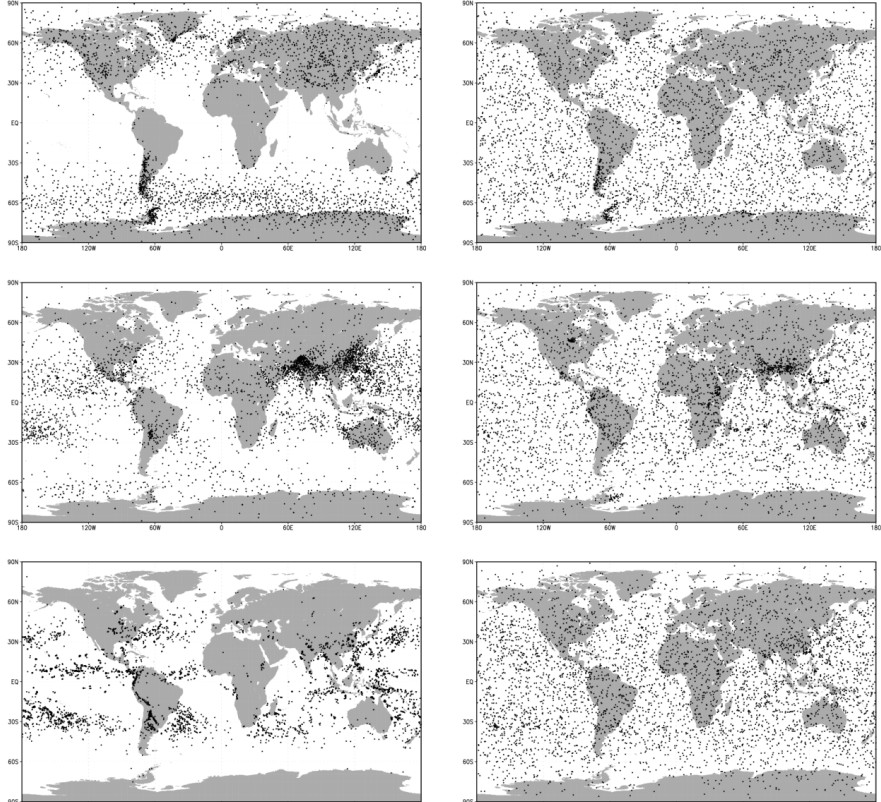

**Figure 8.** Locations of selected profiles in the temperature (top),

specific humidity (middle), and cloud condensate (bottom), sampled

subsets of the IFS-91 (left) and IFS-137 (right) databases (from

https://www.nwpsaf.eu/site/update-137-level-nwp-profile-dataset/ ,

2019).

The temporal distribution of the selected profiles is illustrated in

Fig. 9. Again, the lack of randomized selection results in large

variations from one month to the next in the case of the IFS-91





database (left panel). The different distributions come mainly from
variations in the ozone subset (green parts of each column).
Dominance of randomly-selected profiles in the IFS-137 database
leaves little room for monthly variation in the data count (right
panel). Moreover, the IFS-91 database also supports the mode with
input parameters, such as detection angle, 2 m temperature, cloud
information, and so on. Therefore, it is feasible to use the selected
samples in a statistical multiple regression experiment.

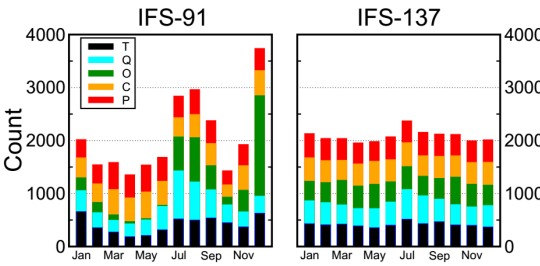

**Figure 9.** Distribution of profiles within the calendar months in
IFS-91 (left) and IFS-137 (right) databases. Different subsets are
shown in different colors.Black parts stand for temperature. Blue
parts represent specific humidity. Green parts indicate ozone subset.
Orange parts stand for cloud condensate. Red parts represent
precipitation.  (from
https://www.nwpsaf.eu/site/update-137-level-nwp-profile-dataset/ ,

2019).


**4.2 Experimental scheme**





In order to verify the retrieval effectiveness of ICS, 5000
temperature profiles provided by the IFS-137 were used for
statistical inversion comparison experiments. The steps are as
follows:
(1) 5000 profiles and their corresponding surface factors,
including surface air pressure, surface temperature, 2 m temperature,
2 m specific humidity, 10 m wind speed, etc. are put into the RTTOV
mode. Then, the AIRS observation brightness temperature is
obtained.
(2) The retrieval of temperature is carried out in accordance with
Eq. (23). The 5000 profiles are divided into two groups. The first
group of 2500 profiles is used to obtain the regression coefficient,
and the second group of 2500 is used to test the result.
(3) Verification of the results. The test is carried out based on the
standard deviation between the retrieval value and the true value.

**4.3 Results and Discussion**
For the statistical inversion comparison experiments, the standard
deviation of temperature retrieval is shown in Fig. 10. First, because
PCS does not take channel sensitivity as a function of height into
consideration, the retrieval result of PCS is inferior to that of ICS.
Second, by comparing the results of ICS and NCS we found that



below 100 hPa, since the method used in this paper considers near
ground to be less of an influencing factor, the channel combination
of ICS is slightly inferior to that of NCS, but the difference is small.
From 100 hPa to10 hPa, the retrieval temperature of ICS in this
paper is consistent with that of NCS, slightly better than the channel
selected for NCS. From 10 hPa to 0.02 hPa, near the space layer, the
retrieval temperature of ICS is obviously better than that of NCS. In
terms of the standard deviation, the channel combination of ICS is
slightly better than that of PCS from 100 hPa to 10 hPa. From 10
hPa to 0.02 hPa, the standard deviation of ICS is lower than that of
NCS at about 1 K, meaning that the retrieval result of ICS is better
than that of NCS.
In order to further illustrate the effectiveness of ICS, the mean
improvement value of the ICS and its percentages compared with the
PCS and NCS in different height are shown in Table 3. Because PCS
does not take channel sensitivity as a function of height into
consideration, the retrieval result of PCS is inferior to that of ICS. In
general, the accuracy of the retrieval temperature of ICS is improved.
Especially, from 100 hPa to 0.01 hPa, the mean value of ICS is
evidently improved by more than 0.5 K which means the accuracy
can be improved by more than 11%. By comparing the results of ICS
and NCS we found that below 100 hPa, since the method used in this





paper considers near ground to be less of an influencing factor, the
channel combination of ICS is slightly inferior to that of NCS, but
the difference is small. From 100 hPa to 0.01 hPa, the mean value of
ICS is improved by more than 0.36 K which means the accuracy can
be improved by more than 9.6%.

**Table 3.** The mean improvement value of the ICS and its
percentages compared with the PCS and NCS in different height.

| Pressure | Improved mean value /Percentage compared with PCS | Improved value /Percentage compared with NCS |
|---|---|---|
| hPa | K/% | K/% |
| surface-100hPa | 0.24/10.77% | -0.04/-3.27% |
| 100hPa-10hPa | 0.15/5.08% | 0.06/2.4% |
| 10hPa-1hPa | 0.04/0.64% | 0.17/2.99% |
| 1hPa-0.01hPa | 0.52/11.92% | 0.36/9.57% |


This is because, as shown in Fig. 4: (1) Near space (20–100 km) is
less affected by the ground surface, so the retrieval result of PCS is
better than that of NCS. (2) Due to the method selected in this paper,
there are more channels at 4.2 μm for $N_2O$ and 4.3 μm for $CO_2$
absorption bands, and the channel combination of PCS is superior to
that of NCS for atmospheric temperature detection in the high
temperature zone. Moreover, ICS takes channel sensitivity as a
function of height into consideration, so its retrieval result is
impressive.




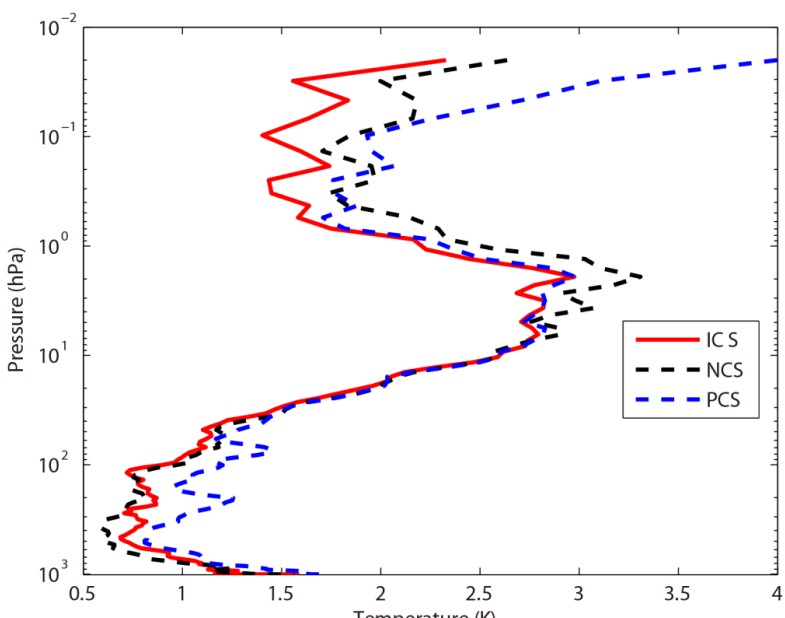


**Figure 10.** The temperature profile standard deviation of statistical

inversion comparison experiments. Red line indicates the result of

ICS. Black dotted line stands for the result of NCS. Blue dotted line

represents the result of PCS.

**5 Statistical inversion comparison experiments in four typical**

**regions**

The accuracy of the retrieval temperature varies from place to place

and changes with weather conditions. Therefore, in order to further

compare the inversion accuracy under different atmospheric

conditions, the atmospheric profile is from the IFS-137 database



introduced in Sect. 4, and divides it into four regions: equatorial
zone, subtropical region, mid-latitude region and Arctic. These
regions' profiles can represent the global typical atmospheric
temperature profiles. The average temperature profiles in these four
regions are shown in Fig. 11. The retrieval temperature varies from
place to place and changes with weather conditions

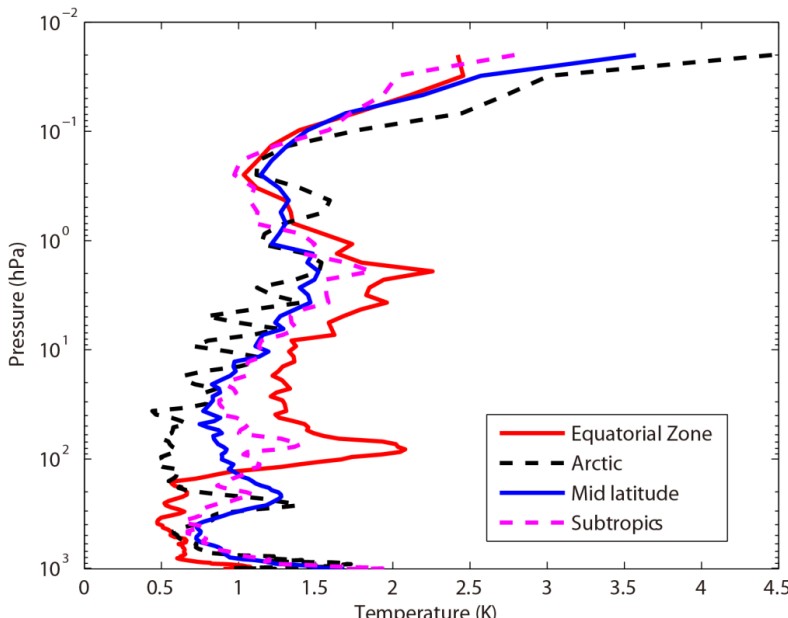


**Figure 11.** The average temperature profiles in four typical regions.
Red line indicates the equatorial zone. Pink dotted line stands for the
subtropics. Blue dotted line represents the mid-latitude region. Black
dotted line stands for the Arctic.






## 5.1 Experimental scheme


In order to further illustrate the different accuracy of the retrieval

temperature using our improved channel selection method under

different atmospheric conditions, the profiles in four typical regions

were used for statistical inversion comparison experiments. The

experimental steps are as follows:

(1) 2500 profiles in Sect. 4 are used to work out the regression

coefficient.

(2) The atmospheric profiles of the four typical regions: equatorial

zone, subtropical region, mid-latitude region and Arctic are used for

statistical inversion comparison experiments and test the result.(3)

Verification of the results. The test is carried out based on the

standard deviation between the retrieval value and the true value.


## 5.2 Results and Discussion


Using statistical inversion comparison experiments in four typical

regions, the standard deviation of temperature retrieval is shown in

Fig. 12. Generally, the retrieval temperature by ICS is greatly

superior to that of NCS and PCS. In particular, above 1 hPa (the near

space layer), the standard deviation of atmospheric temperature can

be optimized to 1 K with PCS and NCS. Thus, ICS shows a great





improvement. The results were consistent with Sect. 4.

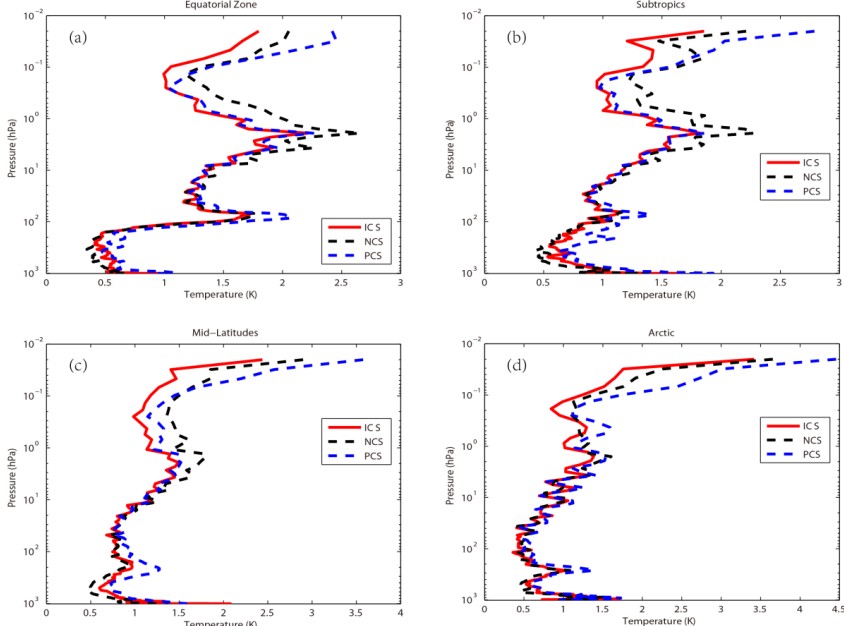


**Figure 12.** The temperature profile standard deviation of statistical

inversion comparison experiments in four typical regions. Red line

indicates the result of ICS. Black dotted line stands for the result of

NCS. Blue dotted line represents the result of PCS. (a) Equatorial

zone. (b) Subtropics. (c) Mid-latitudes. (b) Arctic.


In order to further compare the regional differences of inversion
accuracy, the temperature standard deviation of ICS in four typical
regions are compared in Fig. 13.

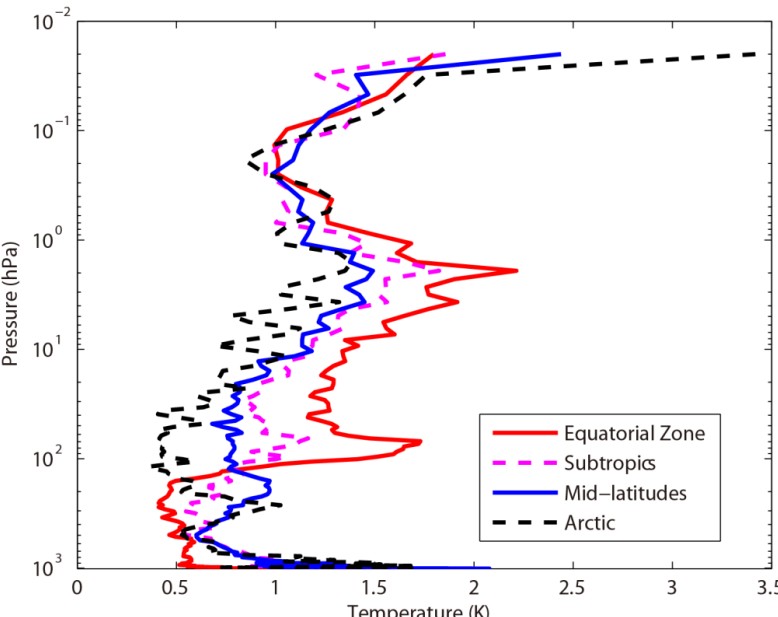


**Figure 13.** The temperature standard deviation of ICS in four typical

regions. Red line indicates the result of equatorial zone. Pink dotted

line represents the result of Subtropics.    Blue line represents the

result of Mid-latitudes. Black dotted line stands for the result of

Arctic.

As can be seen from Fig. 13, the temperature standard deviations

of the ICS in the four typical regions are large. Below100 hPa, due

to the high temperature in the equatorial zone, the channel

combination of ICS is superior to that of PCS and NCS for

atmospheric temperature detection in the high temperature zone. The

standard deviation is 0.5K. Due to the method selected in this paper





there are more channels at 4.2 μm for $N_2O$ and 4.3 μm for $CO_2$
absorption bands which has been previously described in Sect. 3.
Near the tropopause, the standard deviation of the equatorial zone
increases sharply. It is also due to the sharp drops in temperature.
However, the standard deviation of the Arctic is still around 0.5K.
From 100hPa to 1hPa, the standard deviation of ICS is 0.5 K to 2K.
With the increase of latitude, the effectiveness considerably
increases. According to Fig. 12, ICS takes channel sensitivity as a
function of height into consideration, so its retrieval result is
impressive.

In order to further illustrate the effectiveness of ICS, the mean

improvement value of the ICS and its percentages compared with the
PCS and NCS in different height of four typical regions are shown in
Table 4 to Table 7.

**Table 4.** The mean improvement value of the ICS and its
percentages compared with the PCS and NCS in different height in
equatorial zone.

| Pressure | Improved mean value /Percentage compared with PCS | Improved value /Percentage compared with NCS |
|---|---|---|
| hPa | K/% | K/% |
| surface-100hPa | 0.18/12.25% | -0.06/-5.61% |
| 100hPa-10hPa | 0.13/4.23% | 0.04/1.28% |
| 10hPa-1hPa | 0.03/0.09% | 0.24/6.24% |





| 1hPa-0.01hPa | 0.24/7.41% | 0.33/11.22% |


**Table 5.** The mean improvement value of the ICS and its percentages compared with the PCS and NCS in different height in subtropics.

| Pressure | Improved mean value /Percentage compared with PCS | Improved value /Percentage compared with NCS |
|---|---|---|
| hPa | K/% | K/% |
| surface-100hPa | 0.26/12.49% | -0.08/-5.94% |
| 100hPa-10hPa | 0.08/3.55% | 0.02/1.28% |
| 10hPa-1hPa | 0.02/0.56% | 0.2/5.94% |
| 1hPa-0.01hPa | 0.25/7.73% | 0.34/12.51% |


**Table 6.** The mean improvement value of the ICS and its percentages compared with the PCS and NCS in different height in mid-latitudes.

| Pressure | Improved mean value /Percentage compared with PCS | Improved value /Percentage compared with NCS |
|---|---|---|
| hPa | K/% | K/% |
| surface-100hPa | 0.18/9.23% | -0.13/-7.41% |
| 100hPa-10hPa | 0.06/3.68% | 0.03/1.84% |
| 10hPa-1hPa | 0.03/1.03% | 0.18/6.01% |
| 1hPa-0.01hPa | 0.36/10.64% | 0.36/12.71% |


**Table 7.** The mean improvement value of the ICS and its percentages compared with the PCS and NCS in different height in Arctic.





| Pressure | Improved mean value /Percentage compared with PCS | Improved value /Percentage compared with NCS |
|---|---|---|
| hPa | K/% | K/% |
| surface-100hPa | 0.12/6.52% | -0.05/-3.47% |
| 100hPa-10hPa | 0.08/6.59% | 0.02/1.97% |
| 10hPa-1hPa | 0.09/3.64% | 0.06/2.5% |
| 1hPa-0.01hPa | 0.49/13.72% | 0.18/6.47% |


Although the improvements of ICS in the four typical regions are
different, in general, the accuracy of the retrieval temperature of ICS
is improved. Because PCS does not take channel sensitivity as a
function of height into consideration, the retrieval result of PCS is
inferior to that of ICS. In general, the accuracy of the retrieval
temperature of ICS is improved. Especially, from 100 hPa to 0.01
hPa, the accuracy of ICS can be improved by 7% to 13%. By
comparing the results of ICS and NCS we found that below 100 hPa,
since the method used in this paper considers near ground to be less
of an influencing factor, the channel combination of ICS is slightly
inferior to that of NCS, but the difference is small. From 100 hPa to
0.01 hPa, the accuracy of ICS can be improved by 7% to 13%.

**6. Conclusions and discussion**
**6.1 Conclusions**
An improved channel selection method is proposed, based on



information content in this paper. A robust channel selection scheme
and method are proposed, and a series of channel selection
comparison experiments are conducted. The results are as follows:
(1) Since ICS takes channel sensitivity as a function of height into
consideration, the ARI of PCS only tends to be 0.38 and is not
convergent. However, as the $100^{th}$ iteration is approached, the ARI of
ICS tends to be stable, reaching 0.54, while the distribution of the
temperature weight function is more continuous and closer to that of
the actual atmosphere. Thus, in terms of the ARI, convergence, and
the distribution of the temperature weight function, ICS is superior
to PCS.
(2) Statistical inversion comparison experiments show that the
retrieval temperature of ICS in this paper is consistent with that of
NCS. In particular, from 10 hPa to 0.02 hPa (the near space layer),
the retrieval temperature of ICS is obviously better than that of NCS
at about 1 K. In general, the accuracy of the retrieval temperature of
ICS is improved. Especially, from 100 hPa to 0.01 hPa, the accuracy
of ICS can be improved by more than 11%. The reason is that near
space (20–100 km) is less affected by the ground surface, so the
retrieval result of ICS is better than that of NCS. Additionally, due to
the method selected in this paper there are more channels at 4.2 μm
for the $N_2O$ and 4.3 μm for the $CO_2$ absorption bands; the channel



combination of ICS is superior to that of NCS for atmospheric
temperature detection in the high temperature zone.
(3) Statistical inversion comparison experiments in four typical
regions indicate that ICS in this paper is significantly better than
NCS and PCS in different regions and shows latitudinal variations.
Especially, from 100 hPa to 0.01 hPa, the accuracy of ICS can be
improved by 7% to 13%, which means the ICS method selected in
this paper is feasible and shows great promise for applications.

**6.2 Discussion**

In recent years, the atmospheric layer in the altitude range of about
20–100 km has been named "the near space layer" by aeronautical
and astronautical communities. It is between the space-based satellite
platform and the aerospace vehicle platform, which is the transition
zone between aviation and aerospace. Its unique resource has
attracted a lot of attention from many countries. Research and
exploration, therefore, on and of the near space layer are of great
importance. A new channel selection scheme and method for
hyperspectral atmospheric infrared sounder AIRS data based on
layering are proposed. The retrieval results of ICS concerning the
near space atmosphere are particularly good. Thus, ICS aims to
provide a new and an effective channel selection method for the



study of the near space atmosphere using the hyperspectral
atmospheric infrared sounder.

*Data availability.* The data used in this paper are available from the
corresponding author upon request.

*Author contributions.* ZS contributed the central idea. SC, ZS and
HD conceived the method, developed the retrieval algorithm and
discussed the results. SC analyzed the data, prepared the figures and
wrote the paper. WG contributed to refining the ideas, carrying out
additional analyses. All co-authors reviewed the paper.

*Competing interests.* The authors declare that they have no conflict
of interest.

*Acknowledgements.* The study was supported by the National
Natural Science Foundation of China (Grant no. 41875045). The
study was also partly supported by the National Key Research
Program of China: Development of high-resolution data assimilation
technology and atmospheric reanalysis data set in East Asia
(Research on remote sensing telemetry data assimilation technology,
Grant no. 2017YFC1501802).





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
