# Peer review of "Channel selection method for hyperspectral atmospheric infrared sounder using AIRS data based on layering"

_Atmospheric Measurement Techniques, 2019_

## Referee Comment (RC1) · Anonymous Referee #2 · 26 Aug 2019

General comments: The paper proposes an effective channel selection method for a hyperspectral atmospheric infrared sounder using AIRS data based on layering. The newly developed method is compared with two different channel selection methods and provides there an improvement in the retrieval temperature at about 1 K between 20 – 100 km. Statistical inversion comparison experiments based on AIRS data indicate that the standard deviation of atmospheric temperature can be optimized. The authors gave examples for different regions and explained the improvement of the new channel selection method. They explain the importance of the atmospheric layer in the altitude range 20 – 100 km and the importance of an improved channel selection method for this region.

[Figure]

Specific comments: - Explain the abbreviations ICS, NCS and PCS in the abstract. - p. 21, l. 429: Include the following citation: Saunders, R., Hocking, J., Turner, E., Rayer, P., Rundle, D., Brunel, P., Vidot, J., Roquet, P., Matricardi, M., Geer, A., Bormann, N., and Lupu, C.: An update on the RTTOV fast radiative transfer model (currently at versionÂă12), Geosci. Model Dev., 11, 2717–2737, https://doi.org/10.5194/gmd-11-2717-2018, 2018. - p. 21, l. 429-430: The sentence "RTTOV is an evaluation of RRTOV v11, adding and upgrading many features" should be removed because explained is only the common procedure. - Table 1 and table 2 should be removed or shifted to the appendix. - p. 44, l. 747-748: Description of figure is too universal. Please specify the behavior in more detail.

Technical comments: - p.3, l. 65: Atmospheric Infrared Sounder → Atmospheric InfraRed Sounder - p. 14, l. 302: bright → brightness - p. 16, l. 327: bright → brightness - p 26, l. 495: add last access date - p. 29, l. 528: bright → brightness - p. 38, l. 644: add last access date - p. 39, l. 656: Do not write and so on. Either specify the variables or stop the sentence after cloud information. - p. 39, l. 664: add last access date - p. 40, l. 674: Do not write etc. Either specify the variables or stop the sentence after wind speed. - p. 51, l. 872: at 4.3 $\mu$m

---

## Author Comment (AC1) · 28 Aug 2019

Dear Referee #2, We thank you for your review of our manuscript and your detailed remarks. We would like to improve the article here in an immediate reply. Please, find our answers/comments on your notes below:

(1) Explain the abbreviations ICS, NCS and PCS in the abstract. Sorry for my carelessness. In abstract: "In general, the accuracy of the retrieval temperature of ICS is improved. Especially, from 100 hPa to 0.01 hPa, the accuracy of ICS can be improved by more than 11 %; (3) Statistical inversion comparison experiments in four typical regions indicate that ICS in this paper is significantly better than NCS and PCS

[Figure]

in different regions and shows latitudinal variations." (L 31-L 37) This has been modified to "In general, the accuracy of the retrieval temperature of ICS (Improved Channel Selection) is improved. Especially, from 100 hPa to 0.01 hPa, the accuracy of ICS can be improved by more than 11 %; (3) Statistical inversion comparison experiments in four typical regions indicate that ICS in this paper is significantly better than NCS (NWP Channel Selection) and PCS (Primary Channel Selection) in different regions and shows latitudinal variations." (L 31-L 37)

(2) p. 21, L. 429: Include the following citation: Saunders, R., Hocking, J., Turner, E., Rayer, P., Rundle, D., Brunel, P., Vidot, J., Roquet, P., Matricardi, M., Geer, A., Bormann, N., and Lupu, C.: An update on the RTTOV fast radiative transfer model (currently at version 12), Geosci. Model Dev., 11, 2717–2737, https://doi.org/10.5194/gmd-11-2717-2018, 2018. Yes, you are right. This has been added. "For the radiative transfer model and its weight function matrix, K, the RTTOV v12 fast radiative transfer model is used (Saunders et al., 2018)." (L 429-L 430) "Saunders, R., Hocking, J., Turner, E., Rayer, P., Rundle, D., Brunel, P., Vidot, J., Roquet, P., Matricardi, M., Geer, A., Bormann, N., and Lupu, C.: An update on the RTTOV fast radiative transfer model (currently at version 12), Geosci. Model Dev., 11, 2717-2737, https://doi.org/10.5194/gmd-11-2717-2018, 2018." (L 1033-L 1037)

(3) p. 21, L. 429-430: The sentence "RTTOV is an evaluation of RTTOV v11, adding and upgrading many features" should be removed because explained is only the common procedure. Yes, you are right. The sentence "RTTOV is an evaluation of RTTOV v11, adding and upgrading many features" has been removed. (L 429-L 430)

(4) Table 1 and table 2 should be removed or shifted to the appendix. Yes, we agree with you. The two tables have been shifted to the Appendix A. (L 894-L 904)

(5) p. 44, L. 747-748: Description of figure is too universal. Please specify the behavior in more detail. Sorry about this. We explain this as follows. "In order to further compare the regional differences of inversion accuracy, the temperature standard deviations of

ICS in four typical regions are compared in Sect. 5.2." (L 738-L 740)

(6) p. 3, L. 65: Atmospheric Infrared Sounder -> Atmospheric InfraRed Sounder. Yes, we agree with you. This has been modified. (L 66)

(7) p. 14, L. 302: bright-> brightness. Yes, you are right. This has been modified. (L 303)

(8) p. 16, L. 327: bright->brightness. Yes, you are right. This has been modified. (L 328)

(9) p 26, L. 495: add last access date. Yes, we agree with you. "The error covariance matrix of the background, Sa, is calculated using 5000 samples of the IFS-137 data provided by the ECMWF dataset. The last access date is April 26th, 2019 (download address: https://www.nwpsaf.eu/site/update-137-level-nwp-profile-dataset/, 2019)." has been added to "The error covariance matrix of the background, Sa, is calculated using 5000 samples of the IFS-137 data provided by the ECMWF dataset. The last access date is April 26th, 2019 (download address: https://www.nwpsaf.eu/site/update-137-level-nwp-profile-dataset/, 2019)." (L 487)

(10) p. 39, L. 656: Do not write and so on. Either specify the variables or stop the sentence after cloud information. Yes, you are right. "Moreover, the IFS-91 database also supports the mode with input parameters, such as detection angle, 2 m temperature, cloud information, and so on." has been modified to "Moreover, the IFS-91 database also supports the mode with input parameters, such as detection angle, 2 m temperature, cloud information." (L 646)

(11) p. 39, L. 664: add last access date. Yes, we agree with you. "Red parts represent precipitation. (from https://www.nwpsaf.eu/site/ update-137-level-nwp-profile-dataset/ , 2019)." has been added to "Red parts represent precipitation. The last access date is April 26th, 2019. (from https://www.nwpsaf.eu/site/update-137-level-nwp-profile-dataset/ , 2019)." (L 653)
(12) p. 40, L. 674: Do not write etc. Either specify the variables or stop the sentence after wind speed. Yes, you are right. "5000 profiles and their corresponding surface factors, including surface air pressure, surface temperature, 2 m temperature, 2 m specific humidity, 10 m wind speed, etc." has been modified to "5000 profiles and their corresponding surface factors, including surface air pressure, surface temperature, 2 m temperature, 2 m specific humidity, 10 m wind speed." (L 664)

(13) p. 51, l. 872: at 4.3 $\mu$m. Yes, you are right. "...and 4.3 $\mu$m for the CO2 absorption bands;" has been modified to "...and at 4.3 $\mu$m for the CO2 absorption bands;" (L 865)

Thanks again for your careful review. Hopefully our response can enable a further review of the manuscript. Many thanks for your work so far and best regards, Shujie Chang and Co-authors.

Please also note the supplement to this comment:
https://www.atmos-meas-tech-discuss.net/amt-2019-183/amt-2019-183-AC1-supplement.zip

---

## Short Comment (SC1) · 30 Aug 2019

A spectral study method based on AIRS ultra-high spectral resolution atmospheric detection instrument data is a good research paper. In order to improve atmospheric detection capabilities, some countries and organizations have loaded such instruments on their own satellites in recent years, such as IASI/METOP in Europe, CrIS/NPOESS in the US, GIIRS and HIRAS/FY in China. This type of instrument greatly improves the atmospheric detection capability. It has thousands of spectral channels, distribution dependence of height, and there is a correlation between the channels. There are many redundant information, which makes the atmospheric parameter inversion time

too long and unstable. In order to improve the aging time without losing the accuracy of the inversion, it is important to select the spectral channel reasonably. In recent years, many scholars have invested in research in this area and have achieved some gratifying results. The characteristics of this paper are based on the observational information content of satellite instruments. The AIRS data is used to propose an improved effective spectral channel selection method for atmospheric temperature profile inversion. The experimental verification shows that the distribution of the weight function of the spectral channel is continuous, and the temperature accuracy of the statistical method inversion is significantly improved compared with other methods, especially the improvement of the accuracy of the upper atmosphere. The method proposed in this paper is feasible and is conducive to promoting the wide application of ultra-high spectral data.

---

## Short Comment (SC2) · 30 Aug 2019

Dear Mengqingbin, Thank you for your comment of our manuscript and your remarks. Best regards, Shujie Chang
* * *

---

## Short Comment (SC3) · 20 Sep 2019

General comments: This work investigates an effective channel selection scheme for a hyperspectral atmospheric infrared sounder using AIRS data based on layering. Findings from the analysis are important for applications. The retrieval results of ICS concerning the near space atmosphere are particularly good. It is a clearly written paper. At the same time, the article can be improved in several ways as below.
 Technical comments: 1. If the section 2.3 is for methodology, then please add the name to the title of section 2; if not, then move it to the suitable position. 2. There are some grammatical and typographical errors in the manuscript, which I have tried to capture in the

[Figure]

technical comments section. These must be corrected before the manuscript can be published. 414-416: There are a few with extremely large measurement errors, which reduce the accuracy of prediction to some extent. -> Among them, some extremely large measurement errors educe the accuracy of prediction to some extent. 509-513: Therefore, when we select channels, the results differ because of the different observation angles. But due to the selection principle and method are exactly the same and our key is the selection method; we do not discuss, therefore, the variation in observation angle when making a selection. -> The goal of this section is focusing on the selection methods of selecting channels, therefore the biases produced from different observation angles can be ignored. 579: reaching -> and reach to 689-692: "Second, by comparing the results of ICS and NCS we found that below 100 hPa, since the method used in this paper considers near ground to be less of an influencing factor, the channel combination of ICS is slightly inferior to that of NCS, but the difference is small." This sentence is rather awkward, please rewrite. 742-743: the atmospheric profile is from the IFS-137 database introduced in Sect. 4, and divides it into four regions -> this paper divided the atmospheric profile from the IFS-137 database introduced in Sect. 4 into four regions 744-745: These regions' profiles -> The profiles of these regions 798-799: As can be seen from Fig. 13, the temperature standard deviations of the ICS in the four typical regions are large. -> The temperature standard deviations of the ICS in the four typical regions are large (Fig. 13).
* * *

---

## Author Comment (AC2) · 20 Sep 2019

Dear Deming Kong, We thank you for your detailed remarks. We would like to improve the article here in an immediate reply. Please, find our answers/comments on your notes below:

If the section 2.3 is for methodology, then please add the name to the title of section 2; if not, then move it to the suitable position. Yes, you are right. This has been added. "2 Channel selection indicator, scheme and method" (L 173) 414-416: There are a few with extremely large measurement errors, which reduce the accuracy of prediction to some extent. -> Among them, some extremely large measurement errors educe the accuracy of prediction to some extent. Yes, we agree with you. "There are a few with extremely large measurement errors, which reduce the accuracy of prediction to some extent." has been modified to "Among them, some extremely large measurement errors reduce" (L 415-L 417) 509-513: Therefore, when we select channels, the results differ because of the different observation angles. But due to the selection principle and method are exactly the same and our key is the selection method; we do not discuss, therefore, the variation in observation angle when making a selection. -> The goal of this section is focusing on the selection methods of selecting channels; therefore the biases produced from different observation angles can be ignored. Yes, we agree with you. "Therefore, when we select channels, the results differ because of the different observation angles. But due to the selection principle and method are exactly the same and our key is the selection method; we do not discuss, therefore, the variation in observation angle when making a selection." has been modified to "The goal of this section is focusing on the selection methods of selecting channels; therefore the biases produced from different observation angles can be ignored." (L 502-L 504) 579: reaching -> and reach to Yes, you are right. "reaching" has been modified to "and reach to" (L 570). 689-692: "Second, by comparing the results of ICS and NCS we found that below 100 hPa, since the method used in this paper considers near ground to be less of an influencing factor, the channel combination of ICS is slightly inferior to that of NCS, but the difference is small." This sentence is rather awkward, please rewrite. Sorry about this. We think this sentence can express what we what to show the readers, it is not be rewrited. 742-743: the atmospheric profile is from the IFS-137 database introduced in Sect. 4, and divides it into four regions -> this paper divided the atmospheric profile from the IFS-137 database introduced in Sect. 4 into four regions Yes, you are right. "the atmospheric profile is from the IFS-137 database introduced in Sect. 4, and divides it into four regions" has been modified to "this paper divided the atmospheric profile from the IFS-137 database introduced in Sect. 4 into four regions" (L 730-L 731). 744-745: These regions' profiles -> The profiles of these regions Yes, you are right. "These regions' profiles" has been modified to "The profiles of these regions" (L 732-L 733). 798-799: As can be seen from Fig. 13, the temperature standard deviations of the ICS in the four typical regions are large. -> The temperature standard deviations of the ICS in the four typical regions are large (Fig. 13). Yes, you are right. "As can be seen from Fig. 13, the temperature standard deviations of the ICS in the four typical regions are large." has been added to "The temperature standard deviations of the ICS in the four typical regions are large (Fig. 13)." (L 789-L 790).

Thanks again for your careful remark. Many thanks for your work so far and best regards, Shujie Chang and Co-authors. Please also note the supplement to this comment.

Please also note the supplement to this comment:
https://www.atmos-meas-tech-discuss.net/amt-2019-183/amt-2019-183-AC2-supplement.pdf

**Supplement:**

[revised manuscript text omitted]

$$\text{T} = T^* \cdot A, \qquad\qquad\qquad (9)$$

$$T_b = T_b^* \cdot A, \qquad\qquad\qquad (10)$$

where $T^*$ and $T_b^*$ are the eigenvectors of the covariance matrix of temperature (or humidity) and brightness temperature, respectively. A and B stand for the corresponding expansion coefficient vectors of temperature (humidity) and brightness temperature.

Using the least squares method and the orthogonal property, the coefficient conversion matrix, V, is introduced:

$$\text{A} = \text{V} \cdot B, \qquad\qquad\qquad (11)$$

where $V = AB^T(BB^T)^{-1}$. $\qquad\qquad\qquad (12)$

Using the orthogonality, we get:

$$\text{B} = (T_b^*)^T T_b, \qquad\qquad\qquad (13)$$

$$\text{A} = (T^*)^T T. \qquad\qquad\qquad (14)$$

[revised manuscript text omitted]

---

## Referee Comment (RC2) · Anonymous Referee #4 · 4 Nov 2019

General Comments

The paper by Chang et al. introduces a channel selection method for infrared hyperspectral sounders. A particular feature of the method proposed here is that it takes the height dependencies of the kernel functions into account. It is shown that this approach reduces AIRS temperature retrieval errors compared to other methods.

The study has interesting findings, but I found the presentation quality of the manuscript to be rather poor. I would like to ask the authors to please carefully address the numerous specific comments and technical corrections, so that the paper becomes suitable for publication.

[Figure]

Specific Comments

l1-3: I would like to suggest to streamline the title a bit, e.g., "A channel selection method for hyperspectral atmospheric infrared sounders based on layering". Perhaps the reference to the AIRS instrument can be neglected in the title as the method will be applicable to other instruments, too?

l22-24: The statement "The distribution of the temperature weight function is more continuous, more closely approximating that of the actual atmosphere" is unclear to me. Do you mean the coverage or sensitivity of the weighting functions is more evenly distributed over height with this method?

l28: The term "near space layer" is not commonly used, I think. I would suggest to rephrase it by "stratosphere and mesosphere" here and in other places of the manuscript, for clarity.

l31-35: The acronyms ICS, NCS, and PCS need to be introduced in the abstract.

l50-55: The word "detection" is frequently used in the manuscript, but considering that you are referring to temperature, I would recommend to change this to "sounding", "observation", or "measurement" in most instances.

l61-62: Can you add a more recent reference regarding "today's needs" of NWP? Eyre et al. (1993) is more than 20 years old.

l70-72: IASI became operational in 2007 and not in 2010.

l129-130 and l140-142: Which instruments are you referring to here? AIRS has more than 2000 channels.

l147: I would suggest to rephrase "weight function" by "weighting function" here and throughout the manuscript.

l147-148: The statement "... use only the weight function to study appropriate numerical methods, the use of which allows sensitive channels to be selected." is unclear to

me. Please rephrase.

l155-161: The concept of information content itself does consider all the height dependencies of the kernel matrix K (Rodgers, 2000 or Eq. (1) in the present manuscript). Earlier work may have neglected the height dependencies of K for simplicity and to ease the calculations. These sentences should be rephrased so that they do not give the wrong impression that the information content ignores the height dependencies of the weighting functions in general.

l195-195: This is redundant and can be deleted.

l216-219: This sentence is unclear and should be rephrased.

l221-227: The "sequential absorption method" has been described elsewhere before, e.g., (Dudhia et al., 2002):

Dudhia, A., Jay, V. L., & Rodgers, C. D. (2002). Microwindow selection for high-spectral-resolution sounders. Applied Optics, 41(18), 3665-3673.

l231: The expression "\partial^2\Omega / \partial\nu^2" has not been explained and the derivative looks wrong, maybe skip it here?

l262-263: The sentence "According to S_a, S_\epsilon...can be calculated" is unclear and should be rephrased.

l271-275: I would suggest to remove the phrase "... but it still satisfies the optimum value in a certain sense". The method cannot find the global optimum as it applies only a sequential search strategy. It is good to point out this limitation, no need to oversell the results.

l283-285: I did not understand how you are actually making use of the different channel selections for the different heights in the retrieval process. Does this mean the different channel sets are used to evaluate only certain heights in the retrieved profile?

l330 and 344: I got confused about the notation, what is "\hat T'" referring to?

[Figure]

l379-380: "after retrieval of observations has been complete several times" is unclear to me.

l380-383: How do you deal with non-linearity inherent to atmospheric radiative transfer? Isn't that a major problem for multiple regression?

l387: Add a reference for the AIRS instrument, e.g. (Aumann et al., 2003).

l395: The AIRS footprint size 13.5 km at nadir, please check.

l396-398: 4.3 and 15 micron are used simultaneously for both, temperature and carbon dioxide retrievals, I think.

l398: When you refer to "absolute accuracy", does this include noise as well? Noise should not be included as it counts as "precision".

l399-401: "the four imaging channels of visible/near infrared are always filled" is unclear, please explain or remove.

l406-411: This paragraph can be deleted. Everything was already said in the introduction.

l412: Does the "root-mean square error" include both, accuracy and prediction, in this study?

l416-417: Delete the sentence "Moreover, not all channels possess the same measurement error." This is obvious from the figure.

l415: Please provide a reference or a web link for the RMS errors shown here.

l429: Introduce acronym and provide reference for RTTOV.

l429-430: Delete sentence saying "RTTOV is an evolution of..." as this does not provide really useful information.

l431: What is ATOVS?

l440-443: Table 1 is not needed at all in this manuscript, I think. What is the reader supposed to do with it? Just provide the web link for reference.

l451-452: The phrase "The weight function matrix K ..." can be deleted, as this is clear.

l465: What is H(X0)? Is it needed for anything here?

l473-475: Which "products" are you referring to? Is there a reference for the NWPSAF requirements?

l492-495: At this point the reader does not know anything about the IFS-137 data set. It is introduced much later in the manuscript.

l500: What is causing the off-diagonal bands in the temperature covariance matrix? For instance, why is the temperature at 100 hPa closely correlated with temperature at about 60-70 hPa?

l516: The color bar unit is "K^2", but shouldn't this be "K/K", as it refers to a change of brightness temperature with respect to atmospheric temperature, i.e., dBT/dT? (same for Fig. 7)

l532-544: Wavenumbers are missing the unit "cm^-1". Replace "11 micron" by "10 micron".

l546: The term "high temperature zone" is used here and elsewhere in the manuscript, but what does it refer to? Please explain.

l575-576: "Moreover, the temperature profile of each layer can be retrieved." How can you retrieve a profile for a layer? Please clarify.

l569 and 583: Figs. 5 and 6 could be combined in a single figure to allow for a comparison.

l596-605: You are describing the PCS distribution as "scattered", but I would rather apply this word to the ICS distribution. The ICS distributions seems to jump or scatter

from one iteration to the next. The PCS distribution looks much more continuous along the x-axis.

l604-605: Sorry, but I don't know what you mean by "scenario in the real atmosphere" in this context?

l615-619: Please provide a reference for the IFS-137 data set here.

l620-623: Please replace the number of model grid points by something more meaningful such as horizontal resolution of the data. The list of the 137 individual pressure levels and Table 2 are not really needed, I think.

l647-664: As you are not going to make any use of the IFS-91 data set in this study, there is no need to introduce it and discuss the differences with respect to the IFS-137 data set. Section 4.1 could be shortened significantly, I think.

l641: The axes labels in Fig. 8 are too small.

l658: Fig. 9 is not really needed, as it was already pointed out in the text that the coverage of the IFS-137 data set is rather homogeneous.

l674: Why does the RTTOV model need 10 m wind speeds for the radiative transfer calculations?

l744-746: Did you also look at the southern hemisphere polar regions? The sentence "These regions' profiles can represent the global typical atmospheric temperature profiles" makes no sense to me because the regional means are different from the global mean. Delete this.

l746-748: I got very concerned about the mean temperature profiles shown in Fig. 11. If these are regional means of hundreds to thousands of profiles, how can they show wave oscillations and look that noisy? Shouldn't the mean profiles be rather smooth? Is this due to the original selection of the IFS-137 profiles focusing on cases with strong temperature gradients?

l774-775 and 813: It is okay to say that ICS works "better" than NCS or PCS, but saying it is "greatly superior" or "impressive" is overselling the results, I think. Suggest to rephrase this and use more moderate wording.

l819-834: Tables 4 to 7 are largely redundant and can be removed from the paper, I think.

l849 and 882: Suggest to simply delete the headings for Sects. 6.1 and 6.2, as they appear in the wrong order. The conclusions should follow the discussion.

Technical Corrections

l30: remove "evidently"

l38-39: suggest to rephrase "... is feasible and shows great promise for application" as "... shows potential for future applications"

l44: _the_ Earth's

l67-68: suggest to rephrase "AIRS has 2378 spectral channels with subpoint at 13 km and a detection height from the ground of up to 65 km" as "AIRS has 2378 spectral channels providing sensitivity from the ground to up to about 65 km of altitude"

l73: change "attaches" to "devotes" (or similar)

l76: change "detection" to "observations" (or similar)

l80: change "atmospheric detectors" to "instruments"

l83: delete "intense"

l86: change "general satellite detection instrument" to "typical satellite instruments"

l89: change "the center frequency, bandwidth" to "center frequency and bandwidth"

l96: there is _often_ a close correlation between _the channels_

l106: demands of _simulating_ all the channels

l107: to _properly_ select

l151: ignoring _some_ factors

l183: change "\hat S" to "S_\epsilon"

l186: delete "by hyperspectral data"

l187-188: delete "which comes from the selected channel in hyperspectral data with respect to ..." or rephrase to clarify

l209-210: rephrase to "... combination making the information content..."

l235: change "single" to "scalar"

l242: rephrase to "Since S_a and S_\epsilon are ..."

l248: change "pre-observation error" to "a priori uncertainty"

l275: delete "its"

l288: method_s for_ the ... profile_s_

l297: _numerically_ stable

l302: change "bright temperature" to "brightness temperature" (here and throughout the manuscript)

l303: expanded _as_

l367: Taking a derivative of Eq. (21) with respect to G,...

l387: delete "instrument suite" and change to "is _primarily_ designed"

l415: few _channels_

l428: rephrase to "For the calculation of radiative transfer and the weighting function matrix, K, the RTTOV..."

[Figure]

l434: rephrase to "and _trace_ gas concentration_s_"

l439 and 440: delete "v12"

l459 and 461: change "characteristic" to "variable"

l460: delete "radiation"

l481: selection __ in

l491: rephrase to "...of the AIRS channels"

l510: change "But due to" to "However,"

l518: delete "of ICS"

l563: change to "retrieval of temperature"

l611: was used _for the statistical inversion experiments_.

l675: mode_l_. Then, the _simulated AIRS spectra are_ obtained

l696: delete "obviously"

l704 and 719: _at_ different height_s_

l729: change "impressive" to "improved"

l740: change "weather conditions" to "atmospheric conditions" (also elsewhere in the manuscript)

l743: change "and divides it" to "have been divided"

l777: replace "optimized to" by "improved by"

l785: (_d_) Arctic

l892: _is_ proposed

---

## Author Comment (AC3) · 16 Nov 2019

Dear Referee #4, We thank you for your review of our manuscript and your detailed remarks. We would like to improve the article here in an immediate reply. Please, find our answers/comments on your notes below: (1) L1-3: I would like to suggest to streamline the title a bit, e.g., "A channel selection method for hyperspectral atmospheric infrared sounders based on layering". Perhaps the reference to the AIRS instrument can be neglected in the title as the method will be applicable to other instruments, too? Yes, we

agree with you. The method will be applicable to other instruments. Your suggested title "A channel selection method for hyperspectral atmospheric infrared sounders based on layering" is more proper. This has been modified. (L 1-L 3) (2) L22-24: The statement "The distribution of the temperature weight function is more continuous, more closely approximating that of the actual atmosphere" is unclear to me. Do you mean the coverage or sensitivity of the weighting functions is more evenly distributed over height with this method? Sorry about this. The statement has been rewritten. "The coverage of the weighting functions is more evenly distributed over height with this method and closer to the actual atmosphere" (L 22-L 24) (3) L28: The term "near space layer" is not commonly used, I think. I would suggest to rephrase it by "stratosphere and mesosphere" here and in other places of the manuscript, for clarity. Yes, you are right. "In the near space layer especially" has been modified to "In the stratosphere and mesosphere especially". (L 28) "Near space (20–100 km)" has been modified to "Stratosphere and mesosphere". (L 542; L 696-L 697) "(the near space layer)" has been modified to "(the stratosphere and mesosphere)". (L 751-L752; L 825- L 826) "The reason is that near space (20–100 km) is less affected by the ground surface…"has been modified to "The reason is that stratosphere and mesosphere are less affected by the ground surface…" (L 830-L 832) (4) L31-35: The acronyms ICS, NCS, and PCS need to be introduced in the abstract Sorry for my carelessness. In abstract: "In general, the accuracy of the retrieval temperature of ICS is improved. Especially, from 100 hPa to 0.01 hPa, the accuracy of ICS can be improved by more than 11 %; (3) Statistical inversion comparison experiments in four typical regions indicate that ICS in this paper is significantly better than NCS and PCS in different regions and shows latitudinal variations." (L 31-L 37) This has been modified to "In general, the accuracy of the retrieval temperature of ICS (Improved Channel Selection) is improved; (3) Statistical inversion comparison experiments in four typical regions indicate that ICS in this paper is significantly better than NCS (NWP Channel Selection) and PCS (Primary Channel Selection) in different regions and shows latitudinal variations" (L 31-L 36) (5) L 50-55: The word "detection" is frequently used in the manuscript, but considering that you are

referring to temperature, I would recommend to change this to "sounding", "observation", or "measurement" in most instances. Yes, you are right. This has been modified as follows. "...satellite detection technology has developed rapidly" has been modified to "...satellite observation technology has developed rapidly" (L 40-L 41) "From the perspective of vertical atmospheric detection, satellite instruments are developing rapidly. In their infancy, the traditional infrared detection instruments for detecting atmospheric temperature and moisture profiles ..." has been modified to "From the perspective of vertical atmospheric observation, satellite instruments are developing rapidly. In their infancy, the traditional infrared measurement instruments for detecting atmospheric temperature and moisture profiles ..." (L 47-L 50) "... in terms of detection accuracy ... filter-based spectroscopic detection instrument, therefore, ...To meet this challenge, ...for the creation of high-spectral resolution atmospheric detection instruments ..." has been modified to "... in terms of observation accuracy ... filter-based spectroscopic measurement instrument, therefore, ...To meet this challenge,... for the creation of high-spectral resolution atmospheric measurement instruments..." ( L 55-L 61) "... such advanced detection technologies. ... techniques of hyperspectral resolution atmospheric detection." has been modified to "... such advanced sounding technologies. ... techniques of hyperspectral resolution atmospheric observations." (L 72-L 75) "detection data" has been modified to "observation data" (L 80) "...the general satellite detection instrument" has been modified to "...the typical satellite instruments" (L 85) "With the development of detection technology..."has been modified to "With the development of measurement technology..." (L 88-L 89) "temperature detection" has been modified to "temperature observation" (L 391; L 530; L 547; L 701-L 702; L 777-L 778; L 835) (6) L 61-62: Can you add a more recent reference regarding "today's needs" of NWP? Eyre et al. (1993) is more than 20 years old. Sorry for my carelessness. Two recent references have been added to "(Eyre et al., 1993; Prunet et al., 2010; Menzel et al., 2018)" (L 59) "Menzel, W. P., Schmit, T. J., Zhang, P. and Li, J.: Satellite-based atmospheric infrared sounder development and applications, Bull. Amer. Meteor. Soc., 99, 583–603, https://doi.org/10.1175/BAMS-D-16-0293.1, 2018."

(L 974- L 977) "Prunet, P., Thépaut J. N., and Cass, V.: The information content of clear sky IASI radiances and their potential for numerical weather prediction, Q. J. Roy. Meteor. Soc., 124, 211-241, https://doi.org/10.1002/ qj.49712454510, 2010." (L 978- L 981) (7) L 70-72: IASI became operational in 2007 and not in 2010. Yes, you are right. This has been modified to "The United States and Europe, in 2010 and in 2007, also installed the CRIS (Cross-track Infrared Sounder) and the IASI (Inter-Attractive Atmospheric Sounding Interferometer) on polar-orbiting satellites". (L 68-L 71) (8) L129-130 and L140-142: Which instruments are you referring to here? AIRS has more than 2000 channels. Yes, you are right. AIRS has more than 2000 channels. We are sorry for the confusion. This has been modified as follows. "Kuai et al. (2010) analyzed both the Shannon information content and degrees of freedom in channel selection when retrieving CO2 concentrations using thermal infrared remote sensing and indicated that 40 channels could contain 75% of the information from the total of 1016 channels." has been modified to "Kuai et al. (2010) analyzed both the Shannon information content and degrees of freedom in channel selection when retrieving CO2 concentrations using thermal infrared remote sensing and indicated that 40 channels could contain 75% of the information from the total channels." (L 125-L 129) "Richardson et al. (2018) selected 75 from 853 channels using information content analysis to retrieve the cloud optical depth, cloud properties, and position." has been modified to "Richardson et al. (2018) selected 75 from 853 channels based on the high spectral-resolution oxygen A-band instrument on NASA's Orbiting Carbon Observatory-2 (OCO-2), using information content analysis to retrieve the cloud optical depth, cloud properties, and position." (L 138-L 142) (9) L147: I would suggest to rephrase "weight function" by "weighting function" here and throughout the manuscript. Yes, we agree with you. "weight function" has been modified to "weighting function" throughout the manuscript. (10) L 147-148: The statement "... use only the weight function to study appropriate numerical methods, the use of which allows sensitive channels to be selected." is unclear to me. Please rephrase. Sorry for this. "Today's main methods for channel selection (such as the data precision matrix method (Menke, 1984), singular value decomposition method (Prunet

et al., 2010; Zhang et al., 2011; Wang et al., 2014), and the Jacobi method (Aires et al., 1999; Rabier et al., 2010) use only the weight function to study appropriate numerical methods, the use of which allows sensitive channels to be selected." has been modified to "Today's main methods for channel selection use only the weighting function to study appropriate numerical methods, such as the data precision matrix method (Menke, 1984), singular value decomposition method (Prunet et al., 2010; Zhang et al., 2011; Wang et al., 2014), and the Jacobi method (Aires et al., 1999; Rabier et al., 2010). The use of the methods allows sensitive channels to be selected." (L 143-L 149) (11) L 155-161: The concept of information content itself does consider all the height dependencies of the kernel matrix K (Rodgers, 2000 or Eq. (1) in the present manuscript). Earlier work may have neglected the height dependencies of K for simplicity and to ease the calculations. These sentences should be rephrased so that they do not give the wrong impression that the information content ignores the height dependencies of the weighting functions in general. Yes, you are right. "Currently, information content is often employed in channel selection. During retrieval, this method delivers the largest amount of information for the selected channel combination (Rodgers, 1996; Du et al., 2008; He et al., 2012; Richardson et al., 2018). Although this method has made great breakthroughs in both theory and practice, however, it does not take the sensitivity of different channels at different heights into consideration." has been modified to "Currently, information content is often employed in channel selection. During retrieval, this method delivers the largest amount of information for the selected channel combination (Rodgers, 1996; Du et al., 2008; He et al., 2012; Richardson et al., 2018). This method has made great breakthroughs in both theory and practice, and the concept of information content itself does consider all the height dependencies of the kernel matrix K (Rodgers, 2000). However, earlier works have neglected the height dependencies of K for simplicity." (L 155-L 163) (12) L 195-195: This is redundant and can be deleted. . Yes, you are right. The sentence "where $S\_a$ is the error covariance matrix of the background or the estimated value of the atmospheric profile, and S ÌĆ represents the observation error covariance matrix of each hyperspectral detector channel." has been

deleted. (L 195) (13) L 216-219: This sentence is unclear and should be rephrased. Yes, you are right. "Furthermore, under the maximum one p-value, the corresponding channel combination is used as the optimum channel combination; therefore, the entire atmosphere must be calculated M·C_N^n times." has been modified to "Furthermore, there are M layers in the vertical direction of the atmosphere. Therefore, the entire atmosphere must be calculated M·C_N^n times." (L 212-L 214) (14) L 221-227: The "sequential absorption method" has been described elsewhere before, e.g., (Dudhia et al., 2002): Dudhia, A., Jay, V. L., & Rodgers, C. D. (2002). Microwindow selection for high-spectral-resolution sounders. Applied Optics, 41(18), 3665-3673. Yes, you are right. The corresponding references have been cited. It has been modified to "Therefore, it is necessary to design an effective calculation scheme, and such a scheme, i.e., a channel selection method, using iteration is proposed, called the "sequential absorption method" (Dudhia et al., 2002; Du et al., 2008)." (L 216-L 219) "Dudhia, A., Jay, V. L., and Rodgers, C. D.: Microwindow selection for high-spectral-resolution sounders, Appl. Opt. 41, 3665-3673, https://doi.org/10.1364/AO.41.003665, 2002." (L 916-L 918) (15) L 231: The expression "\partialËȨ2\Omega / \partial\nuËȨ2" has not been explained and the derivative looks wrong, maybe skip it here? Yes, you are right. The expression " s_ïĄě $(\partial^2 \Omega)/(\partial\nu^2)$ " has not been explained, which is an unimportant item and can be skipped. The sentence "A diagonal element, s_ïĄě $(\partial^2 \Omega)/(\partial\nu^2)$, in the S_ïĄě matrix is the error variance in the channel." has been deleted. (L 227) (16) L 262-263: The sentence "According to S_a, S_\epsilon...can be calculated" is unclear and should be rephrased. Sorry about this. "According to S_a, S_ïĄě , K and Eq. (6), R, which is r corresponding to all the selected channels, can be calculated." has been modified to "According to S_a, S_ïĄě , K and Eq. (6), R can be calculated." (L 258) (17) L 271-275: I would suggest to remove the phrase "... but it still satisfies the optimum value in a certain sense". The method cannot find the global optimum as it applies only a sequential search strategy. It is good to point out this limitation, no need to oversell the results. Yes, you are right. The sentence "Of course, the combination selected by this method is not completely equivalent to the channel combination

corresponding to the optimum value of C_Nˆn p, but it still satisfies the optimum value in a certain sense." has been removed. (L 265) (18) L 283-285: I did not understand how you are actually making use of the different channel selections for the different heights in the retrieval process. Does this mean the different channel sets are used to evaluate only certain heights in the retrieved profile? Yes, you are right. This has been explained "In this way, different channel sets can be used to evaluate corresponding height in the retrieved profiles." (L 276-L 278) (19) L 330 and 344: I got confused about the notation, what is "\hat T'" referring to? Sorry for my carelessness. "where T ÌĚ and (T_b ) ÌĚ are the corresponding average values of the elements, respectively. Tˆ' and T_bˆ' represent the corresponding anomalies of the elements, respectively." has been modified to "where T ÌĆ stands for the retrieval atmospheric temperature. T ÌĚ and (T_b ) ÌĚ are the corresponding average values of the elements, respectively. T ÌĆˆ' and T_bˆ' represent the corresponding anomalies of the elements, respectively." (L 326-L 329) (20) L 379-380: "after retrieval of observations has been complete several times" is unclear to me. Sorry for the confusion. "It should be noted that the least squares solution obtained here aims to minimize the sum of the error variance for each element in the atmospheric state vector after retrieval of observations has been completed several times." has been modified to "It should be noted that the least squares solution obtained here aims to minimize the sum of the error variance for each element in the atmospheric state vector after retrieval for several times." (L 372-L 374) (21) L 380-383: How do you deal with non-linearity inherent atmospheric radiactive transfer? Isn't that a major problem for multiple regression? Yes, you are right. We have explained this as follows. "Because the statistical inversion method does not directly solve the radiation transfer equation, it has the advantages of fast calculation speed. In addition, the solution is stable, which makes it one of the highest precision methods (Chedin et al., 1985)." (L 288-L 290) This study is aimed to examine the effectiveness of a channel selection method for hyperspectral atmospheric infrared sounders based on layering. The most particular aspect of this work is that it takes the height dependencies of the kernel functions into account. Thus, considering the calculation speed,

the statistical inversion method is used for our channel selection experiment. (22) L 387: Add a reference for the AIRS instrument, e.g. (Aumann et al., 2003). Sorry for my carelessness. The references have been added. "...(Aumann et al., 2003; Hoffmann and Alexander, 2009)" (L 383-L 384) (23) L 395: The AIRS footprint size 13.5 km at nadir, please check. Yes, you are right. The AIRS footprint size 13.5 km at nadir. "The spatial footprint of the infrared channels is 1.1° in diameter, which corresponds to about 15×15 km at the nadir." has been modified to "The footprint size 13.5 km at nadir (Susskind et al., 2003)." (L 389) "Susskind, J., Barnet, C. D. and Blaisdell, J. M.: Retrieval of atmospheric and surface parameters from AIRS/AMSU/HSB data in the presence of clouds, IEEE Trans. Geosci. Remote Sensing, 41, 390-409, https://doi.org/10.1109/TGRS.2002.808236, 2003." (L 1005- L 1008) (24) L 396-398: 4.3 and 15 micron are used simultaneously for both, temperature and carbon dioxide retrievals, I think. Yes, you are right. "The spectral range includes 4.2 $\mu$m for important temperature observation, 15 $\mu$m for CO2, 6.3 $\mu$m for water vapor, and 9.6 $\mu$m for ozone absorption bands." has been modified to "The spectral range includes 4.3 $\mu$m and 15.5 $\mu$m for important temperature observation and CO2, 6.3 $\mu$m for water vapor, and 9.6 $\mu$m for ozone absorption bands (Menzel et al., 2018)." (L 389-L 392) (25) L398: When you refer to "absolute accuracy", does this include noise as well? Noise should not be included as it counts as "precision". Yes, you are right. "The absolute accuracy of the measured radiation is better than 0.2 K ." has been modified to "The root mean square error (RMSE) of the measured radiation is better than 0.2 K (Susskind et al., 2003)". (L 392-L 394) (26) L 399-401: "the four imaging channels of visible/near infrared are always filled" is unclear, please explain or remove. Yes, you are right. The sentence has been removed. (L 394-L 395) (27) L406-411: This paragraph can be deleted. Everything was already said in the introduction. Yes, you are right. This paragraph has been deleted. (L 400) (28) L412: Does the "root-mean square error" include both, accuracy and prediction, in this study? Root mean square error (RMSE) in this study can be described as accuracy in this study. (29) L 416-417: Delete the sentence "Moreover, not all channels possess the same measurement error." This is

obvious from the figure. Yes, you are right. The sentence "Moreover, not all channels possess the same measurement error." has been deleted. (L 404) (30) L 415: Please provide a reference or a web link for the RMS errors shown here. Yes, you are right. "There are a few with extremely large measurement errors, which reduce the accuracy of prediction to some extent." has been modified to "There are a few channels with extremely large measurement errors, which reduce the accuracy of prediction to some extent. Among them, some extremely large measurement errors reduce the accuracy of prediction to some extent. (Susskind et al., 2003)" (L 402-L 406) (31) L 429: Introduce acronym and provide reference for RTTOV. Yes, you are right. "For the radiative transfer model and its weight function matrix, K, the RTTOV v12 fast radiative transfer model is used. RTTOV is an evolution of RTTOV v11, adding and upgrading many features." has been modified to "For the radiative transfer model and its weighting function matrix, K, the RTTOV (Radiative Transfer for TOVS) v12 fast radiative transfer model is used. Although initially developed for the TOVS (TIROS Operational Vertical Sounder) radiometers, RTTOV can now simulate around 90 different satellite sensors measuring in the MW (microwave), IR (infrared) and VIS (visible) regions of the spectrum (Saunders et al., 2018)." (L 417-L 423) (32) L 429-430: Delete sentence saying "RTTOV is an evolution of..." as this does not provide really useful information. Yes, you are right. The sentence has been deleted. (L 423) (33) L 431: What is ATOVS? Sorry for my carelessness. "ATOVS (Advanced TOVS)" has been added. (L 424) (34) L 440-443: Table 1 is not needed at all in this manuscript, I think. What is the reader supposed to do with it? Just provide the web link for reference. Yes, we agree with you. Table 1 and table 2 is not needed. The two tables have been shifted to the Appendix A. (L 846-L 854) (35) L 451-452: The phrase "The weight function matrix K ..." can be deleted, as this is clear. Yes, you are right. The sentence has been deleted. (L 437) (36) L 465: What is H(X0)? Is it needed for anything here? Sorry for the confusion. It has been explained "The RTTOV_K (the K mode), is used to calculate the matrix H(X0) (Eq. (1)) for a given atmospheric profile characteristic." (L 448-L 450) (37) L 473-475: Which "products" are you referring to? Is there a reference for the NWPSAF requirements?

Sorry for this. This has been modified and a reference for the NWPSAF has been added. "NCS is short for NWP channel selection in this paper. The products were released by the NWPSAF 1DVar (one-dimensional variational analysis) scheme, in accordance with the requirements of the NWPSAF." has been modified to "NCS is short for NWP channel selection in this paper. NCS were released by the NWPSAF 1DVar (one-dimensional variational analysis) scheme, in accordance with the requirements of the NWPSAF (Saunders et al., 2018)." (L 456-L 460) (38) L 492-495: At this point the reader does not know anything about the IFS-137 data set. It is introduced much later in the manuscript. Yes, you are right. Because the main idea of this part is to introduce the channel selection experiment, we will introduce IFS-137 data set in Sect. 4. It has been explained "The error covariance matrix of the background, $S\_a$, is calculated using 5000 samples of the IFS-137 data provided by the ECMWF dataset (The detailed information will be introduced in Sect. 4). The last access date is April 26th, 2019 (download address: https://www.nwpsaf.eu/site/update-137-level-nwp-profile-dataset/, 2019)." (L 476-L 482) (39) L 500: What is causing the off-diagonal bands in the temperature covariance matrix? For instance, why is the temperature at 100 hPa closely correlated with temperature at about 60-70 hPa? Yes, you are right. We have noticed the off-diagonal bands in the temperature covariance matrix, especially from 50hPa to 200hPa. There should be some dynamic processes here. But this study is aimed to examine the effectiveness of a channel selection method. As the temperature covariance matrix is the same, so we do not introduce it further here. (40) L 516: The color bar unit is "KËĘ2", but shouldn't this be "K/K", as it refers to a change of brightness temperature with respect to atmospheric temperature, i.e., dBT/dT? (Same for Fig. 7) Sorry for this. This has been modified (Fig. 3: L 500; Fig.6: L 575-L 577) (41) L 532-544: Wavenumbers are missing the unit "cmËĘ-1". Replace "11 micron" by "10 micron". Yes, you are right. The wavenumber unit " cm-1" has been added. (L 518; L 520; L 524; L 527; L 534) "11$\mu$m" has been replaced by "10$\mu$m "(L 517) (42) L 546: The term "high temperature zone" is used here and elsewhere in the manuscript, but what does it refer to? Please explain. Sorry for this. We have added our explana-

tions "Because 4.2 $\mu$m and 4.3 $\mu$m bands are sensitive to high temperature, the higher temperature is, the better observation can be obtained". (L 531-L 533) ". . .in the high temperature zone." has been modified to ". . .to the higher temperature." (L 546-L 547) ". . .the channel combination of ICS is superior to that of PCS and NCS for atmospheric temperature observation in the high temperature zone" has been modified to ". . .the channel combination of ICS is better than that of PCS and NCS for atmospheric temperature observation to the higher temperature" (L 775-L 778; L 834-L 836) (43) L 575-576: "Moreover, the temperature profile of each layer can be retrieved." How can you retrieve a profile for a layer? Please clarify. Sorry for this. After 324 channels are selected for each layer, the temperature profile of each layer can be retrieved based on statistical inversion (see at Sect. 4). "In this paper, the atmosphere is divided into 137 layers, and based on the information content and iteration, 324 channels are selected for each layer. Moreover, the temperature profile of each layer can be retrieved." has been modified to "In this paper, the atmosphere is divided into 137 layers, and based on the information content and iteration, 324 channels are selected for each layer. Then, the temperature profile of each layer can be retrieved based on statistical inversion (see at Sect. 4)" (L 560-L 564) (44) L 569 and 583: Figs. 5 and 6 could be combined in a single figure to allow for a comparison. Yes, we agree with you. Figs.5 and Figs.6 has been combined in a single figure. (Figure 5: L 556) The following figure numbers (Fig. 7-13) have been modified (Fig. 6-12). (45) L 596-605: You are describing the PCS distribution as "scattered", but I would rather apply this word to the ICS distribution. The ICS distributions seems to jump or scatter Thank you for your notice. Sorry for my carelessness. In Fig.6 "(a) PCS. (b) ICS." should be "(a) ICS. (b) PCS." Which has been modified. (L 576-L 577) (46) L 604-605: Sorry, but I don't know what you mean by "scenario in the real atmosphere" in this context? Sorry for the confusion. "(2) regardless of the number of iterations, the maximum value of the weighting function is stable near 300–400 hPa and 600–700 hPa, without scattering, which resembles more closely the scenario in real atmosphere." has been modified to "(2) regardless of the number of iterations, the maximum value of the weighting function is stable near 300–

400 hPa and 600–700 hPa, without scattering, which is closer to the situation in real atmosphere." (L 585-L 588) (47) L 615-619: Please provide a reference for the IFS-137 data set here. Sorry for my carelessness. The corresponding references have been added. "...(Eresmaa and McNally, 2014; Brath et al., 2018)" (L 603) "Brath, M., Fox, S., Eriksson, P., Harlow, R. C., Burgdorf, M., and Buehler, S. A.: Retrieval of an ice water path over the ocean from ISMAR and MARSS millimeter and submillimeter brightness temperatures, Atmos. Meas. Tech., 11, 611–632, https://doi.org/10.5194/amt-11-611-2018, 2018." (L 891-L 895) "Eresmaa, R. and McNally, A. P.: Diverse profile datasets from the ECMWF 137-level short-range forecasts, Tech. rep., ECMWF, 2014." " (L 919-L 921) (48) L 620-623: Please replace the number of model grid points by something more meaningful such as horizontal resolution of the data. The list of the 137 individual pressure levels and Table 2 are not really needed, I think. Yes, we agree with you. "There are two operational analyses each day (at 00z and 12z), and the modeling grid contains 2,140,702 grid points. The pressure levels adopted for IFS-137 are shown in Table 2." has been modified to "There are two operational analyses each day (at 00z and 12z), and approximately 13 000 atmospheric profiles over the ocean. The pressure levels adopted for IFS-137 are shown in Table A2 (see Table A2 in Appendix A)." (L 604- L 608) (49) L 647-664: As you are not going to make any use of the IFS-91 data set in this study, there is no need to introduce it and discuss the differences with respect to the IFS-137 data set. Section 4.1 could be shortened significantly, I think. Yes, we agree with you. "The temporal distribution of the selected profiles is illustrated in Fig. 9. Again, the lack of randomized selection results in large variations from one month to the next in the case of the IFS-91 database (left panel). The different distributions come mainly from variations in the ozone subset (green parts of each column). Dominance of randomly-selected profiles in the IFS-137 database leaves little room for monthly variation in the data count (right panel). Moreover, the IFS-91 database also supports the mode with input parameters, such as detection angle, 2 m temperature, cloud information. Therefore, it is feasible to use the selected samples in a statistical multiple regression experiment." has been modified to "The temporal distribution of the

**AMTD**
[Figure]

selected profiles is illustrated in Fig. 8. The coverage of the IFS-137 data set is more homogeneous than the IFS-91 data set. Moreover, the IFS-137 database supports the mode with input parameters, such as detection angle, 2 m temperature, and cloud information. Therefore, it is feasible to use the selected samples in a statistical multiple regression experiment." (L 628-L 633) (50) L 641: The axes labels in Fig. 8 are too small Sorry for this. It has been modified. (Fig. 7: L 622-L 626) (51) L 658: Fig. 9 is not really needed, as it was already pointed out in the text that the coverage of the IFS-137 data set is rather homogeneous. Because the introduction of IFS-137 data set has been shortened, in order to introduce IFS-137 data set briefly and visually, Fig.8 (previous Fig. 9) can be remained. (52) L 674: Why does the RTTOV model need 10 m wind speeds for the radiative transfer calculations? According to RTTOV users' guide, the new physically-based model (RTTOV) depends on wind speed and skin temperature as well as zenith angle. Some parameters are put into the RTTOV mode. 10 m wind speeds are used to calculate emissivity (details can be seen at RTTOV users' guide). (53) L 744-746: Did you also look at the southern hemisphere polar regions? The sentence "These regions' profiles can represent the global typical atmospheric temperature profiles" makes no sense to me because the regional means are different from the global mean. Delete this. Sorry for my carelessness. We haven't looked at the southern hemisphere polar regions. The sentence "The profiles of these regions can represent the global typical atmospheric temperature profiles." has been deleted. (L 720) (54) L 746-748: I got very concerned about the mean temperature profiles shown in Fig. 11. If these are regional means of hundreds to thousands of profiles, how can they show wave oscillations and look that noisy? Shouldn't the mean profiles be rather smooth? Is this due to the original selection of the IFS-137 profiles focusing on cases with strong temperature gradients? Sorry for my careless. The figure actually is the temperature standard deviation of ICS in four typical regions (Figure 12). I put the wrong figure here. Wave oscillations in this figure is due to the number of profiles in each region are not hundreds to thousands of profiles. For example, in the arctic (80N -90 N), there are 45 samples. So in the figure, wave oscillation is obvious in arctic.

The average temperature profiles in these four regions have been modified (Fig.10). (L 727-L 731) (55) L 774-775 and 813: It is okay to say that ICS works "better" than NCS or PCS, but saying it is "greatly superior" or "impressive" is overselling the results, I think. Suggest to rephrase this and use more moderate wording. Yes, we agree with you. "Generally, the retrieval temperature by ICS is greatly superior to that of NCS and PCS." has been modified to "Generally, the retrieval temperature by ICS is better than that of NCS and PCS." (L 750-L 751) "According to Fig. 12, ICS takes channel sensitivity as a function of height into consideration, so its retrieval result is impressive." has been modified to "According to Fig. 11, ICS takes channel sensitivity as a function of height into consideration, so its retrieval result is better." (L 786-L 788) Other similar problems have been modified. "...the channel combination of PCS is superior to that of NCS for atmospheric temperature observation in the high temperature zone." has been modified to "...the channel combination of PCS is better than that of NCS for atmospheric temperature observation to the higher temperature." (L 546-L 547) "...the ICS method is superior to that of PCS." has been modified to "the ICS method is better than that of PCS." (L 567-L 568) "...the channel combination of ICS is superior to that of PCS and NCS for atmospheric temperature observation in the high temperature zone" has been modified to "...the channel combination of ICS is better than that of PCS and NCS for atmospheric temperature observation to the higher temperature" (L 775-L 778; L 834-L 836) "...ICS is superior to PCS." has been modified to "...ICS is better than PCS." (L 821-L 822) "Moreover, ICS takes channel sensitivity as a function of height into consideration, so its retrieval result is impressive." has been modified to "Moreover, ICS takes channel sensitivity as a function of height into consideration, so its retrieval result is improved." (L 703-L 704) (56) L 819-834: Tables 4 to 7 are largely redundant and can be removed from the paper, I think. Yes, we agree with you. Tables 4 to 7 have been removed from the paper. And the corresponding indication has been modified. (L 795) In abstract "Especially, from 100 hPa to 0.01 hPa, the accuracy of ICS can be improved by more than 11 %; (3) Statistical inversion comparison experiments in four typical regions indicate that ICS in this paper is significantly better than NCS

(NWP Channel Selection) and PCS (Primary Channel Selection) in different regions and shows latitudinal variations. Especially, from 100 hPa to 0.01 hPa, the accuracy of ICS can be improved by 7% to 13%, which means the ICS method selected in this paper is feasible and shows great promise for applications." has been modified to "(3) Statistical inversion comparison experiments in four typical regions indicate that ICS in this paper is significantly better than NCS (NWP Channel Selection) and PCS (Primary Channel Selection) in different regions and shows latitudinal variations, which shows potential for future applications." (L 32-L 36) (57) L 849 and 882: Suggest to simply delete the headings for Sects. 6.1 and 6.2, as they appear in the wrong order. The conclusions should follow the discussion. Yes, you are right. The headings and the correct order have been modified. (L 796-L 840)

Technical Corrections (58) L30: remove "evidently" Yes, you are right. "evidently" has been deleted. (L 30) (59) L 38-39: suggest to rephrase "... is feasible and shows great promise for application" as "... shows potential for future applications" Yes, we agree with you. "(3) Statistical inversion comparison experiments in four typical regions indicate that ICS in this paper is significantly better than NCS (NWP Channel Selection) and PCS (Primary Channel Selection) in different regions and shows latitudinal variations. Especially, from 100 hPa to 0.01 hPa, the accuracy of ICS can be improved by 7% to 13%, which means the ICS method selected in this paper is feasible and shows great promise for applications." has been modified to "(3) Statistical inversion comparison experiments in four typical regions indicate that ICS in this paper is significantly better than NCS (NWP Channel Selection) and PCS (Primary Channel Selection) in different regions and shows latitudinal variations, which shows potential for future applications." (L 32-L 36) "...Especially, from 100 hPa to 0.01 hPa, the accuracy of ICS can be improved by 7% to 13%, which means the ICS method selected in this paper is feasible and shows great promise for applications." has been modified to "..., which shows potential for future applications." (L 839-L 840) (60) L44: _the_ Earth's Yes, you are right "...observe Earth's atmosphere..." has been modified to "...observe the Earth's atmosphere..." (L41) (61) L 67-68: suggest

to rephrase "AIRS has 2378 spectral channels with subpoint at 13 km and a detection height from the ground of up to 65 km" as "AIRS has 2378 spectral channels providing sensitivity from the ground to up to about 65 km of altitude" Yes, you are right. This has been modified. (L 66-L 68) (62) L 73: change "attaches" to "devotes" (or similar) Yes, you are right. This has been modified. (L 72) (63) L 76: change "detection" to "observations" (or similar) Yes, you are right. This has been modified. (L 75) The similar problems have been modified. "...satellite detection technology has developed rapidly" has been modified to "...satellite observation technology has developed rapidly" (L 40-L 41) "From the perspective of vertical atmospheric detection, satellite instruments are developing rapidly. In their infancy, the traditional infrared detection instruments for detecting atmospheric temperature and moisture profiles ..." has been modified to "From the perspective of vertical atmospheric observation, satellite instruments are developing rapidly. In their infancy, the traditional infrared measurement instruments for detecting atmospheric temperature and moisture profiles ..." (L 47-L 50) "... in terms of detection accuracy ... filter-based spectroscopic detection instrument, therefore, ...To meet this challenge, ...for the creation of high-spectral resolution atmospheric detection instruments ..." has been modified to "... in terms of observation accuracy ... filter-based spectroscopic measurement instrument, therefore, ...To meet this challenge,... for the creation of high-spectral resolution atmospheric measurement instruments..." ( L 55-L 61) "... such advanced detection technologies. ... techniques of hyperspectral resolution atmospheric detection." has been modified to "... such advanced sounding technologies. ... techniques of hyperspectral resolution atmospheric observations." (L 72-L 75) "detection data" has been modified to "observation data" (L 80) "...the general satellite detection instrument" has been modified to "...the typical satellite instruments" (L 85) "With the development of detection technology..."has been modified to "With the development of measurement technology..." (L 88-L 89) "temperature detection" has been modified to "temperature observation" (L 391; L 530; L 547; L 701-L 702; L 777-L 778; L 835) (64) L 80: change "atmospheric detectors" to "instruments" Yes, you are right. This

has been modified. (L 79) (65) L 83: delete "intense" Yes, you are right. This has been deleted. (L 82) (66) L 86: change "general satellite detection instrument" to "typical satellite instruments" Yes, you are right. This has been modified. (L 85) (67) L 89: change "the center frequency, bandwidth" to "center frequency and bandwidth" Yes, you are right. This has been modified. (L 88) (68) L 96: there is _often_ a close correlation between _the channels_ Yes, you are right. This has been modified. (L 94-L 95) (70) L 106: demands of _simulating_ all the channels Yes, you are right. This has been modified. (L 105) (71) L 107: to _properly_ select Yes, you are right. This has been modified. (L 106-L 107) (72) L 151: ignoring _some_ factors Yes, you are right. This has been modified. (L 151) (73) L 183: change "\hat S" to "S_\epsilon" Sorry for my careless. "S ÌĆ" has been modified to "S_ïĄě". (L 185) (74) L 186: delete "by hyperspectral data" Yes, you are right. This has been deleted. (L 188) (75) L 187-188: delete "which comes from the selected channel in hyperspectral data with respect to ..." or rephrase to clarify Sorry for this. It has been deleted. (L 188) (76) L 209-210: rephrase to "... combination making the information content..." "This combination make the information content, H, or the ARI defined in this paper as large as possible, in order to maintain the highest possible accuracy in the retrieval results." has been modified to "This combination makes the information content, H, or the ARI defined in this paper as large as possible, in order to maintain the highest possible accuracy in the retrieval results." (L 205-L 208) (77) L 235: change "single" to "scalar" Yes, you are right. This has been modified. (L 230) (78) L 242: rephrase to "Since S_a and S_nepsilon are ..." Yes, you are right. "Since S_a is a positive definite symmetric matrix..." has been modified to "Since S_a and S_ïĄě are positive definite symmetric matrixes..." (L 237) (79) L 248: change "pre-observation error" to "a priori uncertainty" Yes, you are right. This has been modified. (L 243) (80) L 275: delete "its" Yes, you are right. This has been deleted. (L 266) (81) L 288: method_s for_ the ... profile_s_ Yes, you are right. "The inversion method of the atmospheric temperature profile..." has been modified to "The inversion methods for the atmospheric temperature profiles..." (L 281) (82) L 297: _numerically_ stable Yes, you are right. This has been added.

(L 290) (83) L 302: change "bright temperature" to "brightness temperature" (here and throughout the manuscript) Sorry for my careless. Those have been modified. (L 296; L 321; L 503; L 512) (84) L 303: expanded _as_ Yes, you are right. This has been modified. (L 297) (85) L 367: Taking a derivative of Eq. (21) with respect to G,... Yes, you are right. "Equation (21) takes a derivative with respect to G..." has been modified to "Taking a derivative of Eq. (21) with respect to G..." (L 362) (86) L 387: delete "instrument suite" and change to "is _primarily_ designed" Yes, you are right. "The Atmospheric Infrared Sounder (AIRS) instrument suite is designed to..." has been modified to "The Atmospheric Infrared Sounder (AIRS) is primarily designed to..." (L 381) (87) L 415: few _channels_ Yes, you are right. This has been modified. (L 402-L 403) (88) L 428: rephrase to "For the calculation of radiative transfer and the weighting function matrix, K, the RTTOV..." Yes, you are right. This has been modified. (L 417-L418) (89) L 434: rephrase to "and _trace_ gas concentration_s_" Yes, you are right. This has been modified. (L 427) (90) L 439 and 440: delete "v12" Yes, we agree with you. Those have been deleted. (L 431-L 432 and L 433) (91) L 459 and 461: change "characteristic" to "variable" Yes, you are right. This has been modified. (L 443 and L 445) (92) L 460: delete "radiation" Yes, you are right. This has been deleted. (L 444) (93) L 481: selection __ in Yes, you are right. This has been modified. (L 466) (94) L 491: rephrase to "...of the AIRS channels" Yes, you are right. This has been modified. (L 476) (95) L 510: change "But due to" to "However," Yes, we agree with you. "Therefore, when we select channels, the results differ because of the different observation angles. But due to the selection principle and method are exactly the same and our key is the selection method; we do not discuss, therefore, the variation in observation angle when making a selection." has been modified to "The goal of this section is focusing on the selection methods of selecting channels; therefore the biases produced from different observation angles can be ignored." (L 495-L 497) (96) L 518: delete "of ICS" Yes, you are right. This has been deleted. (L 502) (97) L 563: change to "retrieval of temperature" Yes, you are right. This has been modified. (L 550) (98) L 611: was used _for the statistical inversion experiments_. Yes,

you are right. This has been added. (L 594-L 595) (99) L 675: mode_l_. Then, the _simulated AIRS spectra are_ obtained Yes, you are right. This has been modified. (L 651) (100) L 696: delete "obviously" Yes, you are right. This has been deleted. (L 671) (101) L 704 and 719: _at_ different height_s_ Yes, you are right. Those have been modified. (L 679 and L 694) (102) L 729: change "impressive" to "improved" Yes, you are right. This has been modified to "…its retrieval result is improved". (L 704) (103) L 740: change "weather conditions" to "atmospheric conditions" (also elsewhere in the manuscript) Yes, you are right. Those have been modified. (L 715 and L 722) (104) L 743: change "and divides it" to "have been divided" Yes, we agree with you. "…the atmospheric profile is from the IFS-137 database introduced in Sect. 4, and divides it into four regions…" has been modified to "…this paper has divided the atmospheric profile from the IFS-137 database introduced in Sect. 4 into four regions…" (L 716-L 720) (105) L 777: replace "optimized to" by "improved by" Yes, you are right. This has been modified. (L 753) (106) L 785: (_d_) Arctic Sorry for my carelessness. This has been modified. (L 761) (107) L 892: _is_ proposed Yes, you are right. This has been modified. (L 806) Thanks again for your careful review. Hopefully our response can enable a further review of the manuscript. Many thanks for your work so far and best regards, Shujie Chang and Co-authors. Please also note the supplement to this comment.

Please also note the supplement to this comment:
https://www.atmos-meas-tech-discuss.net/amt-2019-183/amt-2019-183-AC3-supplement.zip

---

## Author Comment (AC5) · 18 Nov 2019

Dear Referee #4, We thank you for your review of our manuscript and your detailed remarks. We would like to improve the article here in an immediate reply. You can download the supplement which contains the revised manuscript and the reply. Thanks again.

Please also note the supplement to this comment:
https://www.atmos-meas-tech-discuss.net/amt-2019-183/amt-2019-183-AC5-supplement.zip

---

## Author Response (AR1)

**(a) The point-by-point response to the reviews**

**- Referee #2**

(1) Explain the abbreviations ICS, NCS and PCS in the abstract.

Sorry for my carelessness.

In abstract: "In general, the accuracy of the retrieval temperature of ICS is improved. Especially, from 100 hPa to 0.01 hPa, the accuracy of ICS can be improved by more than 11 %; (3) Statistical inversion comparison experiments in four typical regions indicate that ICS in this paper is significantly better than NCS and PCS in different regions and shows latitudinal variations." (L 31-L 37)

This has been modified to "In general, the accuracy of the retrieval temperature of ICS (Improved Channel Selection) is improved; (3) Statistical inversion comparison experiments in four typical regions indicate that ICS in this paper is significantly better than NCS (NWP Channel Selection) and PCS (Primary Channel Selection) in different regions and shows latitudinal variations, which shows potential for future applications." (L 30-L 36)

Thanks.

(2) p. 21, L. 429: Include the following citation: Saunders, R., Hocking, J., Turner, E., Rayer, P., Rundle, D., Brunel, P., Vidot, J., Roquet, P., Matricardi, M., Geer, A., Bormann, N., and Lupu, C.: An update on the

RTTOV fast radiative transfer model (currently at version 12), Geosci.

Model Dev., 11, 2717–2737, https://doi.org/10.5194/gmd-11-2717-2018,

2018.

Yes, you are right.

This has been added. "…RTTOV can now simulate around 90 different satellite sensors measuring in the MW (microwave), IR (infrared) and

VIS (visible) regions of the spectrum (Saunders et al., 2018)" (L 420-L

423)

"Saunders, R., Hocking, J., Turner, E., Rayer, P., Rundle, D., Brunel, P.,

Vidot, J., Roquet, P., Matricardi, M., Geer, A., Bormann, N., and Lupu,

C.: An update on the RTTOV fast radiative transfer model (currently at version 12), Geosci. Model Dev., 11, 2717-2737, https://doi.org/10.5194/gmd-11-2717-2018, 2018." (L 1000-L 1004)

Thanks.

(3) p. 21, L. 429-430: The sentence "RTTOV is an evaluation of RTTOV

v11, adding and upgrading many features" should be removed because explained is only the common procedure.

Yes, you are right.

The sentence "RTTOV is an evaluation of RTTOV v11, adding and upgrading many features" has been removed. (L 419-L 420)

Thanks.

(4) Table 1 and table 2 should be removed or shifted to the appendix.

Yes, we agree with you.

The two tables have been shifted to the Appendix A. (L 846-L 854)

Thanks.

(5) p. 44, L. 747-748: Description of figure is too universal. Please specify the behavior in more detail.

Sorry about this. We explain this as follows.

"In order to further compare the regional differences of inversion accuracy, the temperature standard deviations of ICS in four typical regions are compared in Sect. 5.2." (L 722-L 724)

Thanks.

(6) p. 3, L. 65: Atmospheric Infrared Sounder -> Atmospheric InfraRed Sounder.

Yes, we agree with you. This has been modified. (L 64)

Thanks.

(7) p. 14, L. 302: bright-> brightness.

Yes, you are right. This has been modified. (L 296; L 321; L 503; L 512)

Thanks.

(8) p. 16, L. 327: bright->brightness.

Yes, you are right. This has been modified. (L 296; L 321; L 503; L 512)

Thanks.

(9) p 26, L. 495: add last access date.

Yes, we agree with you.

"The error covariance matrix of the background, Sa, is calculated using 5000 samples of the IFS-137 data provided by the ECMWF dataset. The last access date is April 26th, 2019 (download address: https://www.nwpsaf.eu/site/update-137-level-nwp-profile-dataset/, 2019)." has been added to "The error covariance matrix of the background, $S_a$, is calculated using 5000 samples of the IFS-137 data provided by the ECMWF dataset (The detailed information will be introduced in Sect. 4). The last access date is April 26th, 2019 (download address: https://www.nwpsaf.eu/site/update-137-level-nwp-profile-dataset/, 2019)." (L 477-L 482)

Thanks.

(10) p. 39, L. 656: Do not write and so on. Either specify the variables or stop the sentence after cloud information.

Yes, you are right.

"Moreover, the IFS-91 database also supports the mode with input parameters, such as detection angle, 2 m temperature, cloud information, and so on." has been modified to "The coverage of the IFS-137 data set is more homogeneous than the IFS-91 data set. Moreover, the IFS-137 database supports the mode with input parameters, such as detection angle, 2 m temperature, and cloud information." (L 629-L 632)

Thanks.

(11) p. 39, L. 664: add last access date.

Yes, we agree with you.

"Red parts represent precipitation.   (from https://www.nwpsaf.eu/site/ update-137-level-nwp-profile-dataset/ , 2019)." has been added to "Red parts represent precipitation. The last access date is April 26th, 2019. (from https://www.nwpsaf.eu/site/update-137-level-nwp-profile-dataset/ , 2019)." (L 638-L 641)

Thanks.

(12) p. 40, L. 674: Do not write etc. Either specify the variables or stop the sentence after wind speed.

Yes, you are right.

 "5000 profiles and their corresponding surface factors, including surface air pressure, surface temperature, 2 m temperature, 2 m specific humidity, 10 m wind speed, etc." has been modified to "5000 profiles and their corresponding surface factors, including surface air pressure, surface temperature, 2 m temperature, 2 m specific humidity, 10 m wind speed." (L 650)

Thanks.

(13) p. 51, l. 872: at 4.3 μm.

Yes, you are right.

"…and 4.3 μm for the $CO_2$ absorption bands;" has been modified to "…and at 4.3 μm for the $CO_2$ absorption bands;" (L 833)

Thanks again for your careful review.

**- Referee #4**

(1) L1-3: I would like to suggest to streamline the title a bit, e.g., "A

channel selection method for hyperspectral atmospheric infrared sounders based on layering". Perhaps the reference to the AIRS instrument can be neglected in the title as the method will be applicable to other instruments, too?

Yes, we agree with you.

The method will be applicable to other instruments. Your suggested title

"A channel selection method for hyperspectral atmospheric infrared sounders based on layering" is more proper.

This has been modified. (L 1-L 3)

Thanks.

(2) L22-24: The statement "The distribution of the temperature weight function is more continuous, more closely approximating that of the actual atmosphere" is unclear to me. Do you mean the coverage or sensitivity of the weighting functions is more evenly distributed over height with this method?

Sorry about this.

The statement has been rewritten. "The coverage of the weighting functions is more evenly distributed over height with this method and closer to the actual atmosphere" (L 22-L 24)

Thanks.

(3) L28: The term "near space layer" is not commonly used, I think. I

  would suggest to rephrase it by "stratosphere and mesosphere" here and

  in other places of the manuscript, for clarity.

Yes, you are right.

"In the near space layer especially" has been modified to "In the stratosphere and mesosphere especially". (L 28)

"Near space (20–100 km)" has been modified to "Stratosphere and mesosphere". (L 542; L 696-L 697)

"(the near space layer)" has been modified to "(the stratosphere and mesosphere)". (L 751-L752; L 825- L 826)

"The reason is that near space (20–100 km) is less affected by the ground surface…"has been modified to "The reason is that stratosphere and mesosphere are less affected by the ground surface…" (L 830-L 832)

Thanks.

(4) L31-35: The acronyms ICS, NCS, and PCS need to be introduced in the abstract

Sorry for my carelessness.

In abstract: "In general, the accuracy of the retrieval temperature of ICS

is improved. Especially, from 100 hPa to 0.01 hPa, the accuracy of ICS

can be improved by more than 11 %; (3) Statistical inversion comparison experiments in four typical regions indicate that ICS in this paper is significantly better than NCS and PCS in different regions and shows latitudinal variations." (L 31-L 37)

This has been modified to "In general, the accuracy of the retrieval temperature of ICS (Improved Channel Selection) is improved; (3) Statistical inversion comparison experiments in four typical regions indicate that ICS in this paper is significantly better than NCS (NWP Channel Selection) and PCS (Primary Channel Selection) in different regions and shows latitudinal variations" (L 31-L 36)

Thanks.

(5) L 50-55: The word "detection" is frequently used in the manuscript, but considering that you are referring to temperature, I would recommend to change this to "sounding", "observation", or "measurement" in most instances.

Yes, you are right. This has been modified as follows.

"…satellite detection technology has developed rapidly" has been modified to "…satellite observation technology has developed rapidly" (L 40-L 41)

"From the perspective of vertical atmospheric detection, satellite instruments are developing rapidly. In their infancy, the traditional infrared detection instruments for detecting atmospheric temperature and moisture profiles …" has been modified to "From the perspective of vertical atmospheric observation, satellite instruments are developing rapidly. In their infancy, the traditional infrared measurement instruments for detecting atmospheric temperature and moisture profiles …" (L 47-L 50)

" … in terms of detection accuracy … filter-based spectroscopic detection instrument, therefore, …To meet this challenge, …for the creation of high-spectral resolution atmospheric detection instruments …" has been modified to "… in terms of observation accuracy … filter-based spectroscopic measurement instrument, therefore, …To meet this challenge,… for the creation of high-spectral resolution atmospheric measurement instruments…" ( L 55-L 61)

" … such advanced detection technologies. … techniques of hyperspectral resolution atmospheric detection." has been modified to "… such advanced sounding technologies. … techniques of hyperspectral resolution atmospheric observations." (L 72-L 75)

"detection data" has been modified to "observation data" (L 80)

"…the general satellite detection instrument" has been modified to "…the typical satellite instruments" (L 85)

"With the development of detection technology…"has been modified to "With the development of measurement technology…" (L 88-L 89)

"temperature detection" has been modified to "temperature observation"

(L 391; L 530; L 547; L 701-L 702; L 777-L 778; L 835)

Thanks.

(6) L 61-62: Can you add a more recent reference regarding "today's needs" of NWP? Eyre et al. (1993) is more than 20 years old.

Sorry for my carelessness.

Two recent references have been added to "(Eyre et al., 1993; Prunet et al., 2010; Menzel et al., 2018)" (L 59)

"Menzel, W. P., Schmit, T. J., Zhang, P. and Li, J.: Satellite-based atmospheric infrared sounder development and applications, Bull. Amer. Meteor. Soc., 99, 583–603, https://doi.org/10.1175/BAMS-D-16-0293.1, 2018." (L 974- L 977)

"Prunet, P., Thépaut J. N., and Cass, V.: The information content of clear sky IASI radiances and their potential for numerical weather prediction, Q. J. Roy. Meteor. Soc., 124, 211-241, https://doi.org/10.1002/qj.49712454510, 2010." (L 978- L 981)

Thanks.

(7) L 70-72: IASI became operational in 2007 and not in 2010.

Yes, you are right.

This has been modified to "The United States and Europe, in 2010 and in 2007, also installed the CRIS (Cross-track Infrared Sounder) and the IASI (Inter-Attractive Atmospheric Sounding Interferometer) on polar-orbiting satellites". (L 68-L 71)

Thanks.

(8) L129-130 and L140-142: Which instruments are you referring to here?

AIRS has more than 2000 channels.

Yes, you are right. AIRS has more than 2000 channels. We are sorry for the confusion. This has been modified as follows.

"Kuai et al. (2010) analyzed both the Shannon information content and degrees of freedom in channel selection when retrieving CO2

concentrations using thermal infrared remote sensing and indicated that

40 channels could contain 75% of the information from the total of 1016

channels." has been modified to "Kuai et al. (2010) analyzed both the

Shannon information content and degrees of freedom in channel selection when retrieving CO2 concentrations using thermal infrared remote sensing and indicated that 40 channels could contain 75% of the information from the total channels." (L 125-L 129)

"Richardson et al. (2018) selected 75 from 853 channels using information content analysis to retrieve the cloud optical depth, cloud properties, and position." has been modified to "Richardson et al. (2018)

selected 75 from 853 channels based on the high spectral-resolution oxygen A-band instrument on NASA's Orbiting Carbon Observatory-2

(OCO-2), using information content analysis to retrieve the cloud optical depth, cloud properties, and position." (L 138-L 142)

Thanks.

(9) L147: I would suggest to rephrase "weight function" by "weighting function" here and throughout the manuscript.

Yes, we agree with you.

"weight function" has been modified to "weighting function" throughout the manuscript.

Thanks.

(10) L 147-148: The statement "... use only the weight function to study appropriate numerical methods, the use of which allows sensitive channels to be selected." is unclear to me. Please rephrase.

Sorry for this.

"Today's main methods for channel selection (such as the data precision matrix method (Menke, 1984), singular value decomposition method (Prunet et al., 2010; Zhang et al., 2011; Wang et al., 2014), and the Jacobi method (Aires et al., 1999; Rabier et al., 2010) use only the weight function to study appropriate numerical methods, the use of which allows sensitive channels to be selected." has been modified to "Today's main methods for channel selection use only the weighting function to study appropriate numerical methods, such as the data precision matrix method (Menke, 1984), singular value decomposition method (Prunet et al., 2010; Zhang et al., 2011; Wang et al., 2014), and the Jacobi method (Aires et al., 1999; Rabier et al., 2010). The use of the methods allows sensitive channels to be selected." (L 143-L 149)

Thanks.

(11) L 155-161: The concept of information content itself does consider all the height dependencies of the kernel matrix K (Rodgers, 2000 or Eq. (1) in the present manuscript). Earlier work may have neglected the height dependencies of K for simplicity and to ease the calculations. These sentences should be rephrased so that they do not give the wrong impression that the information content ignores the height dependencies of the weighting functions in general.

Yes, you are right.

"Currently, information content is often employed in channel selection. During retrieval, this method delivers the largest amount of information for the selected channel combination (Rodgers, 1996; Du et al., 2008; He et al., 2012; Richardson et al., 2018). Although this method has made great breakthroughs in both theory and practice, however, it does not take the sensitivity of different channels at different heights into consideration." has been modified to "Currently, information content is often employed in channel selection. During retrieval, this method delivers the largest amount of information for the selected channel combination (Rodgers, 1996; Du et al., 2008; He et al., 2012; Richardson et al., 2018). This method has made great breakthroughs in both theory and practice, and the concept of information content itself does consider all the height dependencies of the kernel matrix K (Rodgers, 2000). However, earlier works have neglected the height dependencies of K for simplicity." (L

155-L 163)

Thanks.

(12) L 195-195: This is redundant and can be deleted.

. Yes, you are right.

The sentence "where $S_a$ is the error covariance matrix of the background or the estimated value of the atmospheric profile, and $\hat{S}$ represents the observation error covariance matrix of each hyperspectral detector channel."has been deleted. (L 195)

Thanks.

(13) L 216-219: This sentence is unclear and should be rephrased.

Yes, you are right.

"Furthermore, under the maximum one p-value, the corresponding channel combination is used as the optimum channel combination; therefore, the entire atmosphere must be calculated $M \cdot C_N^n$ times." has been modified to "Furthermore, there are M layers in the vertical direction of the atmosphere. Therefore, the entire atmosphere must be calculated $M \cdot C_N^n$ times." (L 212-L 214)

Thanks.

(14) L 221-227: The "sequential absorption method" has been described elsewhere before, e.g., (Dudhia et al., 2002):

Dudhia, A., Jay, V. L., & Rodgers, C. D. (2002). Microwindow selection for high-spectral-resolution sounders. Applied Optics, 41(18), 3665-3673.

Yes, you are right. The corresponding references have been cited.

It has been modified to "Therefore, it is necessary to design an effective calculation scheme, and such a scheme, i.e., a channel selection method, using iteration is proposed, called the "sequential absorption method"

(Dudhia et al., 2002; Du et al., 2008)." (L 216-L 219)

"Dudhia, A., Jay, V. L., and Rodgers, C. D.: Microwindow selection for high-spectral-resolution sounders, Appl. Opt. 41, 3665-3673, https://doi.org/10.1364/AO.41.003665, 2002." (L 916-L 918)

Thanks.

(15) L 231: The expression "\partial^2\Omega / \partial\nu^2" has not been explained and the derivative looks wrong, maybe skip it here?

Yes, you are right.

The expression " $s_\varepsilon \frac{\partial^2 \Omega}{\partial v^2}$ " has not been explained, which is an unimportant item and can be skipped.

The sentence "A diagonal element, $s_\varepsilon \frac{\partial^2 \Omega}{\partial v^2}$, in the $S_\varepsilon$ matrix is the error variance in the channel." has been deleted. (L 227)

Thanks.

(16) L 262-263: The sentence "According to S_a, S_\epsilon...can be calculated" is unclear and should be rephrased.

Sorry about this.

"According to $S_a$, $S_\varepsilon$ , K and Eq. (6), R, which is r corresponding to all the selected channels, can be calculated." has been modified to

"According to $S_a$, $S_\varepsilon$ , K and Eq. (6), R can be calculated." (L 258)

Thanks.

(17) L 271-275: I would suggest to remove the phrase "... but it still satisfies the optimum value in a certain sense". The method cannot find the global optimum as it applies only a sequential search strategy. It is good to point out this limitation, no need to oversell the results.

Yes, you are right.

The sentence "Of course, the combination selected by this method is not completely equivalent to the channel combination corresponding to the optimum value of $C_N^n$ p, but it still satisfies the optimum value in a certain sense." has been removed. (L 265)

Thanks.

(18) L 283-285: I did not understand how you are actually making use of the different channel selections for the different heights in the retrieval process. Does this mean the different channel sets are used to evaluate only certain heights in the retrieved profile?

Yes, you are right.

This has been explained "In this way, different channel sets can be used to evaluate corresponding height in the retrieved profiles." (L 276-L 278)

Thanks.

(19) L 330 and 344: I got confused about the notation, what is "\hat T'"

referring to?

Sorry for my carelessness.

"where $\overline{T}$ and $\overline{T_b}$ are the corresponding average values of the elements, respectively. $T'$ and $T_b'$ represent the corresponding anomalies of the elements, respectively." has been modified to "where $\widehat{T}$ stands for the retrieval atmospheric temperature. $\overline{T}$ and $\overline{T_b}$ are the corresponding average values of the elements, respectively. $\widehat{T}'$ and $T_b'$ represent the corresponding anomalies of the elements, respectively." (L 326-L 329)

Thanks.

(20) L 379-380: "after retrieval of observations has been complete several times" is unclear to me.

Sorry for the confusion.

"It should be noted that the least squares solution obtained here aims to minimize the sum of the error variance for each element in the atmospheric state vector after retrieval of observations has been completed several times." has been modified to "It should be noted that the least squares solution obtained here aims to minimize the sum of the error variance for each element in the atmospheric state vector after retrieval for several times." (L 372-L 374)

Thanks.

(21) L 380-383: How do you deal with non-linearity inherent atmospheric radiactive transfer? Isn't that a major problem for multiple regression?

Yes, you are right. We have explained this as follows.

"Because the statistical inversion method does not directly solve the radiation transfer equation, it has the advantages of fast calculation speed. In addition, the solution is stable, which makes it one of the highest precision methods (Chedin et al., 1985)." (L 288-L 290)

This study is aimed to examine the effectiveness of a channel selection method for hyperspectral atmospheric infrared sounders based on layering. The most particular aspect of this work is that it takes the height dependencies of the kernel functions into account. Thus, considering the calculation speed, the statistical inversion method is used for our channel selection experiment.

Thanks.

(22) L 387: Add a reference for the AIRS instrument, e.g. (Aumann et al., 2003).

Sorry for my carelessness. The references have been added.

"…(Aumann et al., 2003; Hoffmann and Alexander, 2009)" (L 383-L 384)

Thanks.

(23) L 395: The AIRS footprint size 13.5 km at nadir, please check.

Yes, you are right. The AIRS footprint size 13.5 km at nadir.

"The spatial footprint of the infrared channels is 1.1° in diameter, which corresponds to about $15 \times 15$ km at the nadir." has been modified to "The footprint size 13.5 km at nadir (Susskind et al., 2003)." (L 389)

"Susskind, J., Barnet, C. D. and Blaisdell, J. M.: Retrieval of atmospheric and surface parameters from AIRS/AMSU/HSB data in the presence of clouds, IEEE Trans. Geosci. Remote Sensing, 41, 390-409, https://doi.org/10.1109/TGRS.2002.808236, 2003."   (L 1005- L 1008)

Thanks.

(24) L 396-398: 4.3 and 15 micron are used simultaneously for both, temperature and carbon dioxide retrievals, I think.

Yes, you are right.

"The spectral range includes 4.2 µm for important temperature observation, 15 µm for CO2, 6.3 µm for water vapor, and 9.6 µm for ozone absorption bands." has been modified to "The spectral range includes 4.3 µm and 15.5 µm for important temperature observation and

CO2, 6.3 µm for water vapor, and 9.6 µm for ozone absorption bands (Menzel et al., 2018)." (L 389-L 392)

Thanks.

(25)   L398: When you refer to "absolute accuracy", does this include noise as well? Noise should not be included as it counts as "precision".

Yes, you are right.

"The absolute accuracy of the measured radiation is better than 0.2 K ."

has been modified to "The root mean square error (RMSE) of the measured radiation is better than 0.2 K (Susskind et al., 2003)". (L 392-L 394)

Thanks.

(26) L 399-401: "the four imaging channels of visible/near infrared are always filled" is unclear, please explain or remove.

Yes, you are right. The sentence has been removed. (L 394-L 395)

Thanks.

(27) L406-411: This paragraph can be deleted. Everything was already said in the introduction.

Yes, you are right. This paragraph has been deleted. (L 400)

Thanks.

(28) L412: Does the "root-mean square error" include both, accuracy and prediction, in this study?

Root mean square error (RMSE) in this study can be described as accuracy in this study.

Thanks.

(29) L 416-417: Delete the sentence "Moreover, not all channels possess the same measurement error." This is obvious from the figure.

Yes, you are right. The sentence "Moreover, not all channels possess the same measurement error." has been deleted. (L 404)

Thanks.

(30) L 415: Please provide a reference or a web link for the RMS errors shown here.

Yes, you are right.

"There are a few with extremely large measurement errors, which reduce the accuracy of prediction to some extent." has been modified to "There are a few channels with extremely large measurement errors, which reduce the accuracy of prediction to some extent. Among them, some extremely large measurement errors reduce the accuracy of prediction to some extent. (Susskind et al., 2003)" (L 402-L 406)

Thanks.

(31) L 429: Introduce acronym and provide reference for RTTOV.

Yes, you are right.

"For the radiative transfer model and its weight function matrix, K, the

RTTOV v12 fast radiative transfer model is used. RTTOV is an evolution of RTTOV v11, adding and upgrading many features." has been modified to "For the radiative transfer model and its weighting function matrix, K, the RTTOV (Radiative Transfer for TOVS) v12 fast radiative transfer model is used. Although initially developed for the TOVS (TIROS

Operational Vertical Sounder) radiometers, RTTOV can now simulate around 90 different satellite sensors measuring in the MW (microwave),

IR (infrared) and VIS (visible) regions of the spectrum (Saunders et al.,

2018)." (L 417-L 423)

Thanks.

(32) L 429-430: Delete sentence saying "RTTOV is an evolution of..." as this does not provide really useful information.

Yes, you are right. The sentence has been deleted. (L 423)

Thanks.

(33) L 431: What is ATOVS?

Sorry for my carelessness.

"ATOVS (Advanced TOVS)" has been added. (L 424)

Thanks.

(34) L 440-443: Table 1 is not needed at all in this manuscript, I think.

What is the reader supposed to do with it? Just provide the web link for reference.

Yes, we agree with you.

Table 1 and table 2 is not needed.

The two tables have been shifted to the Appendix A. (L 846-L 854)

Thanks.

(35) L 451-452: The phrase "The weight function matrix K ..." can be deleted, as this is clear.

Yes, you are right. The sentence has been deleted. (L 437)

Thanks.

(36) L 465: What is $H(X0)$? Is it needed for anything here?

Sorry for the confusion.

It has been explained "The RTTOV_K (the K mode), is used to calculate the matrix H(X0) (Eq. (1)) for a given atmospheric profile characteristic."

(L 448-L 450)

Thanks.

(37) L 473-475: Which "products" are you referring to? Is there a reference for the NWPSAF requirements?

Sorry for this. This has been modified and a reference for the NWPSAF has been added.

"NCS is short for NWP channel selection in this paper. The products were released by the NWPSAF 1DVar (one-dimensional variational analysis) scheme, in accordance with the requirements of the NWPSAF." has been modified to "NCS is short for NWP channel selection in this paper. NCS were released by the NWPSAF 1DVar (one-dimensional variational analysis) scheme, in accordance with the requirements of the NWPSAF (Saunders et al., 2018)." (L 456-L 460)

Thanks.

(38) L 492-495: At this point the reader does not know anything about the IFS-137 data set. It is introduced much later in the manuscript.

Yes, you are right. Because the main idea of this part is to introduce the channel selection experiment, we will introduce IFS-137 data set in Sect. 4.

It has been explained "The error covariance matrix of the background, $S_a$, is calculated using 5000 samples of the IFS-137 data provided by the

ECMWF dataset (The detailed information will be introduced in Sect. 4).

The last access date is April 26th, 2019 (download address:

https://www.nwpsaf.eu/site/update-137-level-nwp-profile-dataset/, 2019).''

(L 476-L 482)

Thanks.

(39) L 500: What is causing the off-diagonal bands in the temperature covariance matrix? For instance, why is the temperature at 100 hPa closely correlated with temperature at about 60-70 hPa?

Yes, you are right. We have noticed the off-diagonal bands in the temperature covariance matrix, especially from 50hPa to 200hPa. There should be some dynamic processes here. But this study is aimed to examine the effectiveness of a channel selection method. As the temperature covariance matrix is the same, so we do not introduce it further here.

Thanks.

(40) L 516: The color bar unit is "K^2", but shouldn't this be "K/K", as it refers to a change of brightness temperature with respect to atmospheric temperature, i.e., dBT/dT? (Same for Fig. 7)

Sorry for this. This has been modified (Fig. 3: L 500; Fig.6: L 575-L 577)

Thanks.

(41) L 532-544: Wavenumbers are missing the unit "cm^-1". Replace "11

micron" by "10 micron".

Yes, you are right.

The wavenumber unit " cm$^{-1}$" has been added. (L 518; L 520; L 524; L 527; L 534)

"11μm" has been replaced by "10μm "(L 517)

Thanks.

(42) L 546: The term "high temperature zone" is used here and elsewhere in the manuscript, but what does it refer to? Please explain.

Sorry for this. We have added our explanations "Because 4.2 μm and 4.3 μm bands are sensitive to high temperature, the higher temperature is, the better observation can be obtained". (L 531-L 533)

"…in the high temperature zone." has been modified to "…to the higher temperature." (L 546-L 547)

"…the channel combination of ICS is superior to that of PCS and NCS for atmospheric temperature observation in the high temperature zone" has been modified to "…the channel combination of ICS is better than that of PCS and NCS for atmospheric temperature observation to the higher temperature" (L 775-L 778; L 834-L 836)

Thanks.

(43) L 575-576: "Moreover, the temperature profile of each layer can be retrieved." How can you retrieve a profile for a layer? Please clarify.

Sorry for this.

After 324 channels are selected for each layer, the temperature profile of each layer can be retrieved based on statistical inversion (see at Sect. 4).

"In this paper, the atmosphere is divided into 137 layers, and based on the information content and iteration, 324 channels are selected for each layer. Moreover, the temperature profile of each layer can be retrieved." has been modified to "In this paper, the atmosphere is divided into 137 layers, and based on the information content and iteration, 324 channels are selected for each layer. Then, the temperature profile of each layer can be retrieved based on statistical inversion (see at Sect. 4)" (L 560-L 564)

Thanks.

(44) L 569 and 583: Figs. 5 and 6 could be combined in a single figure to allow for a comparison.

Yes, we agree with you.

Figs.5 and Figs.6 has been combined in a single figure. (Figure 5: L 556)

The following figure numbers (Fig. 7-13) have been modified (Fig. 6-12).

Thanks.

(45) L 596-605: You are describing the PCS distribution as "scattered", but I would rather apply this word to the ICS distribution. The ICS distributions seems to jump or scatter

Thank you for your notice. Sorry for my carelessness.

In Fig.6 "(a) PCS. (b) ICS." should be "(a) ICS. (b) PCS." Which has been modified. (L 576-L 577)

Thanks.

(46) L 604-605: Sorry, but I don't know what you mean by "scenario in the real atmosphere" in this context?

Sorry for the confusion.

"(2) regardless of the number of iterations, the maximum value of the weighting function is stable near 300–400 hPa and 600–700 hPa, without scattering, which resembles more closely the scenario in real atmosphere." has been modified to "(2) regardless of the number of iterations, the maximum value of the weighting function is stable near 300–400 hPa and 600–700 hPa, without scattering, which is closer to the situation in real atmosphere." (L 585-L 588)

Thanks.

(47) L 615-619: Please provide a reference for the IFS-137 data set here.

Sorry for my carelessness.

The corresponding references have been added.

"…(Eresmaa and McNally, 2014; Brath et al., 2018)" (L 603)

"Brath, M., Fox, S., Eriksson, P., Harlow, R. C., Burgdorf, M., and Buehler, S. A.: Retrieval of an ice water path over the ocean from ISMAR and MARSS millimeter and submillimeter brightness temperatures, Atmos. Meas. Tech., 11, 611–632, https://doi.org/10.5194/amt-11-611-2018, 2018." (L 891-L 895)

"Eresmaa, R. and McNally, A. P.: Diverse profile datasets from the

ECMWF 137-level short-range forecasts, Tech. rep., ECMWF, 2014."

" (L 919-L 921)

Thanks.

(48) L 620-623: Please replace the number of model grid points by
something more meaningful such as horizontal resolution of the data. The
list of the 137 individual pressure levels and Table 2 are not really needed,
I think.

Yes, we agree with you.

"There are two operational analyses each day (at 00z and 12z), and the
modeling grid contains 2,140,702 grid points. The pressure levels adopted
for IFS-137 are shown in Table 2." has been modified to "There are two
operational analyses each day (at 00z and 12z), and approximately 13 000
atmospheric profiles over the ocean. The pressure levels adopted for
IFS-137 are shown in Table A2 (see Table A2 in Appendix A)." (L 604- L
608)

Thanks.

(49) L 647-664: As you are not going to make any use of the IFS-91 data
set in this study, there is no need to introduce it and discuss the
differences with respect to the IFS-137 data set. Section 4.1 could be
shortened significantly, I think.

Yes, we agree with you.

"The temporal distribution of the selected profiles is illustrated in Fig. 9.

Again, the lack of randomized selection results in large variations from one month to the next in the case of the IFS-91 database (left panel). The different distributions come mainly from variations in the ozone subset (green parts of each column). Dominance of randomly-selected profiles in the IFS-137 database leaves little room for monthly variation in the data count (right panel). Moreover, the IFS-91 database also supports the mode with input parameters, such as detection angle, 2 m temperature, cloud information. Therefore, it is feasible to use the selected samples in a statistical multiple regression experiment." has been modified to "The temporal distribution of the selected profiles is illustrated in Fig. 8. The coverage of the IFS-137 data set is more homogeneous than the IFS-91

data set. Moreover, the IFS-137 database supports the mode with input parameters, such as detection angle, 2 m temperature, and cloud information. Therefore, it is feasible to use the selected samples in a statistical multiple regression experiment." (L 628-L 633)

Thanks.

(50) L 641: The axes labels in Fig. 8 are too small

Sorry for this.

It has been modified. (Fig. 7: L 622-L 626)

Thanks.

(51) L 658: Fig. 9 is not really needed, as it was already pointed out in the text that the coverage of the IFS-137 data set is rather homogeneous.

Because the introduction of IFS-137 data set has been shortened, in order to introduce IFS-137 data set briefly and visually, Fig.8 (previous Fig. 9) can be remained.

Thanks.

(52) L 674: Why does the RTTOV model need 10 m wind speeds for the radiative transfer calculations?

According to RTTOV users' guide, the new physically-based model (RTTOV) depends on wind speed and skin temperature as well as zenith angle. Some parameters are put into the RTTOV mode. 10 m wind speeds are used to calculate emissivity (details can be seen at RTTOV users' guide).

Thanks.

(53) L 744-746: Did you also look at the southern hemisphere polar regions? The sentence "These regions' profiles can represent the global typical atmospheric temperature profiles" makes no sense to me because the regional means are different from the global mean. Delete this.

Sorry for my carelessness.

We haven't looked at the southern hemisphere polar regions. The sentence "The profiles of these regions can represent the global typical atmospheric temperature profiles." has been deleted. (L 720)

Thanks.

(54) L 746-748: I got very concerned about the mean temperature profiles shown in Fig. 11. If these are regional means of hundreds to thousands of profiles, how can they show wave oscillations and look that noisy? Shouldn't the mean profiles be rather smooth? Is this due to the original selection of the IFS-137 profiles focusing on cases with strong temperature gradients?

Sorry for my careless.

The figure actually is the temperature standard deviation of ICS in four typical regions (Figure 12). I put the wrong figure here. Wave oscillations in this figure is due to the number of profiles in each region are not hundreds to thousands of profiles. For example, in the arctic (80N -90 N), there are 45 samples. So in the figure, wave oscillation is obvious in arctic.

The average temperature profiles in these four regions have been modified (Fig.10). (L 727-L 731)

Thanks.

(55) L 774-775 and 813: It is okay to say that ICS works "better"

than NCS or PCS, but saying it is "greatly superior" or "impressive"

is overselling the results, I think. Suggest to rephrase this and use more moderate wording.

Yes, we agree with you.

"Generally, the retrieval temperature by ICS is greatly superior to that of NCS and PCS." has been modified to "Generally, the retrieval temperature by ICS is better than that of NCS and PCS." (L 750-L 751)

"According to Fig. 12, ICS takes channel sensitivity as a function of height into consideration, so its retrieval result is impressive." has been modified to "According to Fig. 11, ICS takes channel sensitivity as a function of height into consideration, so its retrieval result is better." (L 786-L 788)

Other similar problems have been modified.

"…the channel combination of PCS is superior to that of NCS for atmospheric temperature observation in the high temperature zone." has been modified to "…the channel combination of PCS is better than that of NCS for atmospheric temperature observation to the higher temperature." (L 546-L 547)

"…the ICS method is superior to that of PCS." has been modified to "the ICS method is better than that of PCS." (L 567-L 568)

"…the channel combination of ICS is superior to that of PCS and NCS for atmospheric temperature observation in the high temperature zone" has been modified to "…the channel combination of ICS is better than that of PCS and NCS for atmospheric temperature observation to the higher temperature" (L 775-L 778; L 834-L 836)

"…ICS is superior to PCS." has been modified to "…ICS is better than

PCS." (L 821-L 822)

"Moreover, ICS takes channel sensitivity as a function of height into consideration, so its retrieval result is impressive." has been modified to "Moreover, ICS takes channel sensitivity as a function of height into consideration, so its retrieval result is improved." (L 703-L 704)

Thanks.

(56) L 819-834: Tables 4 to 7 are largely redundant and can be removed from the paper, I think.

Yes, we agree with you.

Tables 4 to 7 have been removed from the paper. And the corresponding indication has been modified. (L 795)

In abstract "Especially, from 100 hPa to 0.01 hPa, the accuracy of ICS can be improved by more than 11 %; (3) Statistical inversion comparison experiments in four typical regions indicate that ICS in this paper is significantly better than NCS (NWP Channel Selection) and PCS (Primary Channel Selection) in different regions and shows latitudinal variations. Especially, from 100 hPa to 0.01 hPa, the accuracy of ICS can be improved by 7% to 13%, which means the ICS method selected in this paper is feasible and shows great promise for applications." has been modified to "(3) Statistical inversion comparison experiments in four typical regions indicate that ICS in this paper is significantly better than NCS (NWP Channel Selection) and PCS (Primary Channel Selection) in (57) L 849 and 882: Suggest to simply delete the headings for Sects.

6.1 and 6.2, as they appear in the wrong order. The conclusions should follow the discussion.

Yes, you are right.

The headings and the correct order have been modified. (L 796-L 840)

Thanks.

Technical Corrections (58) L30: remove "evidently"

Yes, you are right.

"evidently" has been deleted. (L 30)

Thanks.

(59) L 38-39: suggest to rephrase "... is feasible and shows great promise for application" as "... shows potential for future applications"

Yes, we agree with you.

 "(3) Statistical inversion comparison experiments in four typical regions indicate that ICS in this paper is significantly better than NCS (NWP

Channel Selection) and PCS (Primary Channel Selection) in different regions and shows latitudinal variations. Especially, from 100 hPa to 0.01

hPa, the accuracy of ICS can be improved by 7% to 13%, which means the ICS method selected in this paper is feasible and shows great promise for applications." has been modified to "(3) Statistical inversion comparison experiments in four typical regions indicate that ICS in this paper is significantly better than NCS (NWP Channel Selection) and PCS (Primary Channel Selection) in different regions and shows latitudinal variations, which shows potential for future applications." (L 32-L 36)

"…Especially, from 100 hPa to 0.01 hPa, the accuracy of ICS can be improved by 7% to 13%, which means the ICS method selected in this paper is feasible and shows great promise for applications." has been modified to "…, which shows potential for future applications." (L 839-L 840)

Thanks.

(60) L44: _the_ Earth's

Yes, you are right

"…observe Earth's atmosphere…" has been modified to "…observe the Earth's atmosphere…" (L41)

Thanks.

(61) L 67-68: suggest to rephrase "AIRS has 2378 spectral channels with subpoint at 13 km and a detection height from the ground of up to 65 km" as "AIRS has 2378 spectral channels providing sensitivity from the ground to up to about 65 km of altitude"

Yes, you are right. This has been modified. (L 66-L 68)

Thanks.

(62) L 73: change "attaches" to "devotes" (or similar)

Yes, you are right. This has been modified. (L 72)

Thanks.

(63) L 76: change "detection" to "observations" (or similar)

Yes, you are right. This has been modified. (L 75)

The similar problems have been modified.

"…satellite detection technology has developed rapidly" has been modified to "…satellite observation technology has developed rapidly" (L 40-L 41)

"From the perspective of vertical atmospheric detection, satellite instruments are developing rapidly. In their infancy, the traditional infrared detection instruments for detecting atmospheric temperature and moisture profiles …" has been modified to "From the perspective of vertical atmospheric observation, satellite instruments are developing rapidly. In their infancy, the traditional infrared measurement instruments for detecting atmospheric temperature and moisture profiles …" (L 47-L 50)

" … in terms of detection accuracy … filter-based spectroscopic detection instrument, therefore, …To meet this challenge, …for the creation of high-spectral resolution atmospheric detection instruments …"

has been modified to "… in terms of observation accuracy … filter-based spectroscopic measurement instrument, therefore, …To meet this challenge,… for the creation of high-spectral resolution atmospheric measurement instruments…" ( L 55-L 61)

"… such advanced detection technologies. … techniques of hyperspectral resolution atmospheric detection." has been modified to "… such advanced sounding technologies. … techniques of hyperspectral resolution atmospheric observations." (L 72-L 75)

"detection data" has been modified to "observation data" (L 80)

"…the general satellite detection instrument" has been modified to "…the typical satellite instruments" (L 85)

"With the development of detection technology…"has been modified to "With the development of measurement technology…" (L 88-L 89)

"temperature detection" has been modified to "temperature observation" (L 391; L 530; L 547; L 701-L 702; L 777-L 778; L 835)

Thanks.

(64) L 80: change "atmospheric detectors" to "instruments"

Yes, you are right. This has been modified. (L 79)

Thanks.

(65) L 83: delete "intense"

Yes, you are right. This has been deleted. (L 82)

Thanks.

(66) L 86: change "general satellite detection instrument" to "typical satellite instruments"

Yes, you are right. This has been modified. (L 85)

Thanks.

(67) L 89: change "the center frequency, bandwidth" to "center frequency and bandwidth"

Yes, you are right. This has been modified. (L 88)

Thanks.

(68) L 96: there is _often_ a close correlation between _the channels_

Yes, you are right. This has been modified. (L 94-L 95)

Thanks.

(69) L 106: demands of _simulating_ all the channels

Yes, you are right. This has been modified. (L 105)

Thanks.

(70) L 107: to _properly_ select

Yes, you are right. This has been modified. (L 106-L 107)

Thanks.

(71) L 151: ignoring _some_ factors

Yes, you are right. This has been modified. (L 151)

Thanks.

(72) L 183: change "\hat S" to "S_\epsilon"

Sorry for my careless.

"Ŝ" has been modified to "$S_\varepsilon$". (L 185)

Thanks.

(73) L 186: delete "by hyperspectral data"

Yes, you are right. This has been deleted. (L 188)

Thanks.

(74) L 187-188: delete "which comes from the selected channel in hyperspectral data with respect to ..." or rephrase to clarify

Sorry for this. It has been deleted. (L 188)

Thanks.

(75) L 209-210: rephrase to "... combination making the information content..."

"This combination make the information content, H, or the ARI defined in this paper as large as possible, in order to maintain the highest possible accuracy in the retrieval results." has been modified to "This combination makes the information content, H, or the ARI defined in this paper as large as possible, in order to maintain the highest possible accuracy in the retrieval results." (L 205-L 208)

Thanks.

(76) L 235: change "single" to "scalar"

Yes, you are right. This has been modified. (L 230)

Thanks.

(77) L 242: rephrase to "Since S_a and S_nepsilon are ..."

Yes, you are right.

"Since $S_a$ is a positive definite symmetric matrix..." has been modified
to "Since $S_a$ and $S_\varepsilon$ are positive definite symmetric matrixes…" (L 237)

Thanks.

(78) L 248: change "pre-observation error" to "a priori uncertainty"

Yes, you are right. This has been modified. (L 243)

Thanks.

(79) L 275: delete "its"

Yes, you are right. This has been deleted. (L 266)

Thanks.

(80) L 288: method_s for_ the ... profile_s_

Yes, you are right.

"The inversion method of the atmospheric temperature profile…" has
been modified to "The inversion methods for the atmospheric
temperature profiles…" (L 281)

Thanks.

(81) L 297: _numerically_ stable

Yes, you are right. This has been added. (L 290)

Thanks.

(82) L 302: change "bright temperature" to "brightness temperature"
(here and throughout the manuscript)

Sorry for my careless. Those have been modified. (L 296; L 321; L 503; L 512)

Thanks.

(83) L 303: expanded _as_

Yes, you are right. This has been modified. (L 297)

Thanks.

(84) L 367: Taking a derivative of Eq. (21) with respect to G,...

Yes, you are right.

"Equation (21) takes a derivative with respect to G…" has been modified to "Taking a derivative of Eq. (21) with respect to G…" (L 362)

Thanks.

(85) L 387: delete "instrument suite" and change to "is _primarily_ designed"

Yes, you are right.

"The Atmospheric Infrared Sounder (AIRS) instrument suite is designed to…" has been modified to "The Atmospheric Infrared Sounder (AIRS) is primarily designed to…" (L 381)

Thanks.

(86) L 415: few _channels_

Yes, you are right. This has been modified. (L 402-L 403)

Thanks.

(87) L 428: rephrase to "For the calculation of radiative transfer and the weighting function matrix, K, the RTTOV..."

Yes, you are right. This has been modified. (L 417-L418)

Thanks.

(88) L 434: rephrase to "and _trace_ gas concentration_s_"

Yes, you are right. This has been modified. (L 427)

Thanks.

(89) L 439 and 440: delete "v12"

Yes, we agree with you. Those have been deleted. (L 431-L 432 and L 433)

Thanks.

(90) L 459 and 461: change "characteristic" to "variable"

Yes, you are right. This has been modified. (L 443 and L 445)

Thanks.

(91) L 460: delete "radiation"

Yes, you are right. This has been deleted. (L 444)

Thanks.

(92) L 481: selection __ in

Yes, you are right. This has been modified. (L 466)

Thanks.

(93) L 491: rephrase to "...of the AIRS channels"

Yes, you are right. This has been modified. (L 476)

Thanks.

(94) L 510: change "But due to" to "However,"

Yes, we agree with you.

"Therefore, when we select channels, the results differ because of the different observation angles. But due to the selection principle and method are exactly the same and our key is the selection method; we do not discuss, therefore, the variation in observation angle when making a selection." has been modified to "The goal of this section is focusing on the selection methods of selecting channels; therefore the biases produced from different observation angles can be ignored." (L 495-L 497)

Thanks.

(95) L 518: delete "of ICS"

Yes, you are right. This has been deleted. (L 502)

Thanks.

(96) L 563: change to "retrieval of temperature"

Yes, you are right. This has been modified. (L 550)

Thanks.

(97) L 611: was used _for the statistical inversion experiments_.

Yes, you are right. This has been added. (L 594-L 595)

Thanks.

(98) L 675: mode_l_. Then, the _simulated AIRS spectra are_ obtained

Yes, you are right. This has been modified. (L 651)

Thanks.

(99) L 696: delete "obviously"

Yes, you are right. This has been deleted. (L 671)

Thanks.

(100) L 704 and 719: _at_ different height_s_

Yes, you are right. Those have been modified. (L 679 and L 694)

Thanks.

(101) L 729: change "impressive" to "improved"

Yes, you are right.

This has been modified to "…its retrieval result is improved". (L 704)

Thanks.

(102) L 740: change "weather conditions" to "atmospheric conditions" (also elsewhere in the manuscript)

Yes, you are right. Those have been modified. (L 715 and L 722)

Thanks.

(103) L 743: change "and divides it" to "have been divided"

Yes, we agree with you.

 "…the atmospheric profile is from the IFS-137 database introduced in

Sect. 4, and divides it into four regions…" has been modified to "…this paper has divided the atmospheric profile from the IFS-137 database introduced in Sect. 4 into four regions…" (L 716-L 720)

Thanks.

(104) L 777: replace "optimized to" by "improved by"

Yes, you are right. This has been modified. (L 753)

Thanks.

(105) L 785: (_d_) Arctic

Sorry for my carelessness. This has been modified. (L 761)

Thanks.

(106) L 892: _is_ proposed

Yes, you are right. This has been modified. (L 806)

Thanks again for your careful review.

**(b) The list of all relevant changes made in the manuscript**

[revised manuscript text omitted]

---

## Author Response (AR2)

**(a) The point-by-point response to the reviews**

Dear Referee #2,

We thank you for your review of our manuscript and your detailed remarks. We would like to improve the article here in an immediate reply.

Please, find our answers/comments on your notes below:

(1) p. 5, l. 108: ...from the thousands of channels' observations... please rephrase, but I have no suggestions

Yes, you are right.

"It is important to properly … from the thousands of channels'

observations …." has been modified to "In order to improve the calculation efficiency and retrieval quality, it is very important to properly select a set of channels that can provide as much information as possible."

(L 104-L 107)

Thanks.

(2) p. 8, l. 155: Channel selection mostly uses the information content and delievers the largest amount of information for the selected channel combination during the retreival.

Yes, you are right.

"Currently, information content is often employed in channel selection.

During retrieval, this method delivers the largest amount of information for the selected channel combination (Rodgers, 1996; Du et al., 2008; He et al., 2012; Richardson et al., 2018)." has been modified to "Channel selection mostly uses the information content and delivers the largest amount of information for the selected channel combination during the retrieval (Rodgers, 1996; Du et al., 2008; He et al., 2012; Richardson et al., 2018)." (L 153-L 156)

Thanks.

(3) p. 9, l. 194: after the retrieval

Yes, you are right.

"…after retrieval" has been modified to "…after the retrieval" (L 193)

Thanks.

(4) p. 11, l. 236: (2) The number looks like an equation number, so it is a bit confusing.

Sorry for the confusion.

The step number has been modified to (I), (II), (III) and (IV). (L 222; L 234; L 255; L 270; L 272)

Thanks.

(5) p. 18, l. 389: The footprint size is 13.5 km

Yes, you are right.

This has been modified. (L 288)

Thanks.

(6) p. 19, l. 401/402: in Fig. 1. The measurement error is not below 0.2K for all the instrument channels. There are...

Yes, you are right.

"The root mean square error of an AIRS infrared channel is shown in Fig. 1, with black spots, indicating that not all the instrument channels possess a measurement error of less than 0.2 K." has been modified to "The root mean square error of an AIRS infrared channel is shown in Fig. 1. The measurement error is not below 0.2K for all the instrument channels." (L 400-L 401)

Thanks.

(7) p. 21, l. 434: ranking -> in the range

Yes, you are right.

This has been modified. (L 432)

Thanks.

(8) p. 27, l. 532: temperature, a better observation can be obtained for higher temperatures

Yes, you are right.

"…temperature, the higher temperature is, the better observation can be obtained;" has been modified to "…temperature, a better observation can be obtained for higher temperatures" (L 524-L 525)

Thanks.

(9) p. 28, l. 547: to the -> at

Yes, you are right.

This has been modified. (L 538)

Thanks.

(10) p. 30, l. 567: reach to -> reaches

Yes, you are right.

This has been modified. (L 560)

Thanks.

(11) p. 36, l. 650: humidity and 10m wind speed (remove dot after speed)

Sorry for my carelessness.

This has been modified. (L 643)

Thanks.

(12) p. 43, l. 778: to the -> at

Yes, you are right.

This has been modified. (L 771; L 828)

Thanks again for your careful review.

Dear Referee #4,

We thank you for your review of our manuscript and your detailed remarks. We would like to improve the article here in an immediate reply.

Please, find our answers/comments on your notes below:

(1) p1, l16-20: Suggest to replace the first two sentences of the abstract by: "This study introduces an effective channel selection method for hyperspectral infrared sounders. The method is illustrated for the

Atmospheric InfraRed Sounder (AIRS) instrument."

Yes, we agree with you.

This has been modified. (L 16-L 18)

Thanks.

(2) p1, l21: "improved method" -> "improved channel selection (ICS)

method"

Yes, you are right.

This has been modified. (L 19-L 21)

Thanks.

(3) p2, l24: suggest to delete "and closer to the actual atmosphere"

Yes, you are right.

This has been deleted. (L 22)

Thanks.

(4) p2, l30: suggest to replace "In general, ..." by "Also at lower heights, ..."

Yes, you are right.

This has been modified. (L 28)

Thanks.

(5) p2, l31-32: delete "(Improved Channel Selection)"

Yes, you are right.

This has been deleted. (L 29)

Thanks.

(6) p2, l32-36: suggest to rephrase this as "Statistical inversion comparison experiments for four different regions illustrate latitudinal and seasonal variations and better performance of ICS compared to the

NWP Channel Selection (NCS) and Primary Channel Selection (PCS)

methods. The ICS method shows potential for future applications."

Yes, we agree with you.

"Statistical inversion comparison experiments in four typical regions indicate that ICS in this paper is significantly better than NCS (NWP Channel Selection) and PCS (Primary Channel Selection) in different regions and shows latitudinal variations, which shows potential for future applications." has been modified to "Statistical inversion comparison experiments for four different regions illustrate latitudinal and seasonal variations and better performance of ICS compared to the NWP Channel Selection (NCS) and Primary Channel Selection (PCS) methods. The ICS method shows potential for future applications." (L 30-L 34)

Thanks.

(7) p4, l70: please fix: "Infrared Atmospheric Sounding Interferometer"

Yes, you are right.

This has been modified. (L 68-L 69)

Thanks.

(8) p5, l110: replace "the channel selection algorithm" by "channel selection algorithms"

Yes, you are right.

This has been modified. (L 108)

Thanks.

(9) p11, l236: replace "p matrix" by "for calculating the p value"?

Yes, you are right.

This has been modified. (L 234)

Thanks.

(10) p11, l237: replace "matrixes, it" by "matrices, they"

Yes, you are right.

This has been modified. (L 235)

Thanks.

(11) p12, l257: replace "According to" by "Using"

Yes, you are right.

This has been modified. (L 257)

Thanks.

(12) p18, l383: The reference to Hoffmann and Alexander (200) can be deleted here, I think. Maybe move reference to Susskind et al. (2003)

from line 389 to this place.

Yes, we agree with you.

"…(Aumann et al., 2003; Hoffmann and Alexander, 2009)." has been modified to "…(Aumann et al., 2003; Susskind et al., 2003).". (L

382-L383)

"Susskind et al. (2003)" has been deleted. (L 388)

Thanks.

(13) p27, l515-534: Suggest to delete all the sentences relating the wavelengths to the wavenumbers, as this is trivial. For instance, replace "When the wavenumber approaches 1000... Near this band..." by "Near the 10 um band..."

Yes, you are right.

"(1) When the wavenumber approaches 1000, the wavelength is 10 μm (1/1000 cm-1). Near this band, fewer channels are selected by PCS because the retrieval of ground temperature is considered by NCS; (2) When the wavenumber is near 1200, the wavelength is 9 μm (1/1200 cm-1). Near this band, no channels are selected by PCS because the retrieval of O3 is not considered in this paper; (3) When the wavenumber approaches 1500, the wavelength is 6.7 μm (1/1500 cm-1). As is known, the spectral range from 6 μm to 7 μm corresponds to water vapor absorption bands, but fewer channels are selected by NCS; (4) When the wavenumber is close to 2000, it derives a wavelength of 5 μm (1/2000 cm-1), which includes 4.2 μm for N2O and 4.3 μm for CO2 absorption bands." has been modified to "(1) Near 10 μm band, fewer channels are selected by PCS because the retrieval of ground temperature is considered by NCS; (2) Near 9 μm band, no channels are selected by PCS because the retrieval of O3 is not considered in this paper; (3) As is known, the spectral range from 6 μm to 7 μm corresponds to water vapor absorption bands, but fewer channels are selected by NCS; (4) Near 5 μm band, it includes 4.2 µm for N2O and 4.3 µm for CO2 absorption bands." (L 514-L 521)

"(5) In the near infrared area, the wavenumber exceeds 2200, deriving a wavelength of less than 4 µm (1/2000 cm-1). A small number of channels is selected by NCS, but no channels are selected by PCS." has been modified to "(5) Near 4 µm band, a small number of channels is selected by NCS, but no channels are selected by PCS." (L 526-L 527)

Thanks.

(14) p27, l529-533: It is still not clear what is meant by "high temperature zone", I think. Does this refer to temperatures at high altitudes?

Sorry for the confusion.

"high temperature zone" is not clear for readers which we want to say "under the higher temperature conditions" But it can be deleted in this sentence, because we have explained it in the following sentences.

"PCS is favorable for atmospheric temperature observation in the high temperature zone. Because 4.2 µm and 4.3 µm bands are sensitive to high temperature, a better observation can be obtained for higher temperatures;" has been modified to "PCS is favorable for atmospheric temperature observation. Because 4.2 µm and 4.3 µm bands are sensitive to high temperature, a better observation can be obtained for higher temperatures;" (L 522-L 525)

"Due to the method selected in this paper, there are more channels at 4.2

μm for N2O and 4.3 μm for CO2 absorption bands, and the channel combination of PCS is superior to that of NCS for atmospheric temperature observation in the high temperature zone." has been modified to "Due to the method selected in this paper, there are more channels at

4.2 μm for N2O and 4.3 μm for CO2 absorption bands, and the channel combination of PCS is superior to that of NCS for atmospheric temperature observation." (L 689-L 693)

Thanks.

(15) p28, l537: replaced "used in this paper" by "considered in this study"

Yes, you are right.

This has been modified. (L 528)

Thanks.

(16) p29, Fig. 5: I would like to suggest to combine the curves of plot a)

and b) into a single plot.

Yes, you are right.

Fig. 5 has been modified to "The relationship between the number of iterations and ARI. Blue line represents the result of ICS. Red dotted line stands for the result of PCS." (L 547: Fig. 5)

Thanks again for your careful review.

(b) **The list of all relevant changes made in the manuscript**

[revised manuscript text omitted]